# Interpretable Concept-Based Memory Reasoning

**David Debot**
KU Leuven
david.debot@kuleuven.be

**Pietro Barbiero**
Universita' della Svizzera Italiana
University of Cambridge
barbiero@tutanota.com

**Francesco Giannini**
Scuola Normale Superiore
francesco.giannini@sns.it

**Gabriele Ciravegna**
DAUIN, Politecnico di Torino
gabriele.ciravegna@polito.it

**Michelangelo Diligenti**
University of Siena
michelangelo.diligenti@unisi.it

**Giuseppe Marra**
KU Leuven
giuseppe.marra@kuleuven.be

## Abstract

The lack of transparency in the decision-making processes of deep learning systems presents a significant challenge in modern artificial intelligence (AI), as it impairs users' ability to rely on and verify these systems. To address this challenge, Concept-Based Models (CBMs) have made significant progress by incorporating human-interpretable concepts into deep learning architectures. This approach allows predictions to be traced back to specific concept patterns that users can understand and potentially intervene on. However, existing CBMs' task predictors are not fully interpretable, preventing a thorough analysis and any form of formal verification of their decision-making process prior to deployment, thereby raising significant reliability concerns. To bridge this gap, we introduce Concept-based Memory Reasoner (CMR), a novel CBM designed to provide a human-understandable and provably-verifiable task prediction process. Our approach is to model each task prediction as a neural selection mechanism over a memory of learnable logic rules, followed by a symbolic evaluation of the selected rule. The presence of an explicit memory and the symbolic evaluation allow domain experts to inspect and formally verify the validity of certain global properties of interest for the task prediction process. Experimental results demonstrate that CMR achieves better accuracy-interpretability trade-offs to state-of-the-art CBMs, discovers logic rules consistent with ground truths, allows for rule interventions, and allows pre-deployment verification.

## 1 Introduction

The opaque decision process of deep learning (DL) systems represents one of the most fundamental problems in modern artificial intelligence (AI). For this reason, eXplainable AI (XAI) [1–3] is currently one of the most active research areas in AI. Among XAI techniques, Concept-Based Models (CBMs) [4–8] represented a significant innovation that made DL models explainable-by-design by introducing a layer of human-interpretable concepts within DL architectures. CBMs consist of at least two functions: a concept encoder, which maps low-level raw features (e.g. an image's pixels) to high-level interpretable concepts (e.g. "red" and "round"), and a task predictor, which uses the learned concepts to solve a downstream task (e.g. "apple"). This way, each task prediction can be traced back

38th Conference on Neural Information Processing Systems (NeurIPS 2024).

to a specific pattern of concepts, thus allowing CBMs to provide explanations in terms of high-level interpretable concepts (e.g. concepts "red" and "round" were both active when the model classified an image as "apple") rather than low-level raw features (e.g. there were 100 red pixels when the model classified an image as "apple"). In other terms, a CBM's task predictor allows understanding *what* the model sees in a given input rather than simply pointing to *where* it is looking [9].

However, state-of-the-art CBMs' task predictors are either unable to solve complex tasks (e.g. linear layers), non-differentiable (e.g. decision trees), or black-box neural networks. CBMs employing black-box task predictors are still considered *locally* interpretable, as concept interventions allow humans to understand how concepts influence predictions for individual input examples. However, they lack *global* interpretability: the human cannot interpret the model's global behaviour, i.e. on any possible instance. This prevents a proper understanding of the model's working, as well as any chance of formally verifying the task predictor decision-making process prior to deployment, thus raising significant concerns in practical applications. As a result, a knowledge gap persists in the existing literature: the definition of a CBM with a task predictor whose behaviour can be inspected, verified, and potentially intervened upon *before* the deployment of the system.

To address this gap, we propose Concept-based Memory Reasoner (CMR), a new CBM where the behaviour and explanations can be inspected and verified before the model is deployed. CMR's task predictor offers global interpretability as it utilizes a differentiable memory of learnable logic rules, making all potential decision rules transparent to humans. Additionally, CMR avoids the concept bottleneck that often limits the accuracy of interpretable models when compared to black-box approaches. Our key innovation lies in an attention mechanism that dynamically selects a relevant rule from the memory, which CMR uses to accurately map concepts to downstream classes.

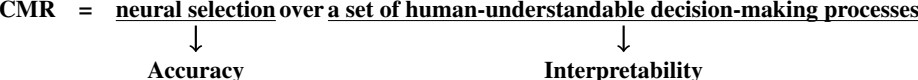

We call this paradigm Neural Interpretable Reasoning (NIR), which involves neurally generating (i.c. selecting from memory) an interpretable model (i.c. a logic rule) and symbolically executing it. Once learned, the memory of logic rules can be interpreted as a disjunctive theory, which can be used for explaining and automatic verification. This verification can take place before the model is deployed and, thus, for any possible input the model will face at deployment time. The concept-based nature of CMR allows the automatic verification of properties that are expressed in terms of high-level human-understandable *concepts* (e.g. "never predict class 'apple' when the concept 'blue' is active") rather than raw features (e.g. "never predict class 'apple' when there are less than ten red pixels").

Our experimental results show that CMR: (i) improves over the accuracy-interpretability performances of state-of-the-art CBMs, (ii) discovers logic rules matching ground truths, (iii) enables rule interventions beyond concept interventions, and (iv) allows verifying properties for their predictions and explanations *before* deployment. Our code is available at `https://github.com/daviddebot/CMR`.

## 2 Preliminary

**Concept Bottleneck Models (CBNMs)** [4, 10] are functions composed of (i) a concept encoder $g : X \to C$ mapping each entity $x \in X \subseteq \mathbb{R}^d$ (e.g. an image) to a set of $n_C$ concepts $c \in C$ (e.g. "red", "round"), and (ii) a task predictor $f : C \to Y$ mapping concepts to the class $y \in Y$ (e.g. "apple") representing a downstream task. For simplicity, in this paper, a single task class is discussed, as multiple tasks can be encoded by instantiating multiple task predictors. When sigmoid activations are used for concepts and task predictions, we can consider $g$ and $f$ as parameterizing a Bernoulli distribution of truth assignments to propositional boolean concepts and tasks. For example, $g_{red}(x) = 0.8$ means that there is an 80% probability for the proposition "$x$ is red" to be true. During training, concept and class predictions $(c, y)$ are aligned with ground-truth labels $(\hat{c}, \hat{y})$. This architecture and training allows CBNMs to provide explanations for class predictions indicating the presence or absence of concepts. Another main advantage of these models is that, at test time, human experts may also *intervene* on mispredicted concept labels to improve CBNMs' task performance and extract counterfactual explanations [4, 11]. However, the task prediction $f$ is still often a black-box model to guarantee high performances, thus not providing any insight into which concepts are used and how they are composed to reach the final prediction.

# 3 Model

In this section, we introduce Concept-based Memory Reasoner (CMR), the first concept-based model that is *globally interpretable*, *provably verifiable* and a *universal binary classifier*. CMR consists of three main components: a concept encoder, a rule selector and a task predictor. CMR's task prediction process differs significantly from traditional CBMs. It operates transparently by (1) selecting a logic rule from a set of jointly-learned rules, and (2) symbolically evaluating the chosen rule on the concept predictions. This unique approach enables CMR not only to provide explanations by tracing class predictions back to concept activations, but also to explain which concepts are utilized and how they interact to make a task prediction. Moreover, the set of learned rules remains accessible throughout the learning process, allowing users to analyse the model's behaviour and automatically verify whether some desired properties are being fulfilled at any time. The logical interpretation of CMR's task predictor, combined with its provably verifiable behaviour, distinguishes it sharply from existing CBMs' task predictors.

## 3.1 Probabilistic graphical model

In Figure 1, we show the probabilistic graphical model of CMR. There are four variables, three of which are standard in (discriminative) CBMs: the observed input $x \in X$, the concepts encoding $c \in C$ and the task prediction $y \in \{0, 1\}$. CMR adds an additional variable:
the rule $r \in \{P, N, I\}^{n_C}$. A rule is a conjunction in the concept set, like $c_1 \wedge \neg c_3$. A conjunction is uniquely identified when, for each concept $c_i$, we know whether, in the rule, the concept is *irrelevant (I)*, *positive (P)* or *negative (N)*. We call $r_i \in \{P, N, I\}$ the role of the $i$-th concept in rule $r$. For example, given the three concepts $c_1, c_2, c_3$, the conjunction $c_1 \wedge \neg c_3$ can be represented as $r_1 = P, r_2 = I, r_3 = N$, since the role of $c_1$ is positive (P), the role of $c_2$ is irrelevant (I) and the role of $c_3$ is negative (N).

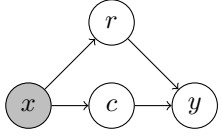

Figure 1: Probabilistic graphical model of CMR

This probabilistic graphical model encodes the joint conditional distribution $p(y, r, c|x)$ and factorizes as

$$p(y, r, c|x) = p(y|c, r)p(r|x)p(c|x) \tag{1}$$

and consists of the following components:

- $p(c|x)$ is the **concept encoder**. For concept bottleneck encoders[1], it is simply the product of $n_C$ independent Bernoulli distributions $p(c_i|x)$, whose logits are parameterized by some neural network encoder $g_i : X \to \mathbb{R}$.

- $p(r|x)$ is the **rule selector**, described in Section 3.1.1. Given an input $x$, $p(r|x)$ models the uncertainty over which conjunctive rule must be used.

- $p(y|c, r)$ is the **task predictor**, which will be described in Section 3.1.2. Given a rule $r \sim p(r|x)$ and an assignment of truth values $c \sim p(c|x)$ to the concepts, the task predictor evaluates the rule on the concepts. In all the cases described in this paper, $p(y|c, r)$ is a degenerate deterministic distribution.

### 3.1.1 Rule selector

We model the rule selector $p(r|x)$ as a mixture of $n_R \in \mathbb{N}$ rule distributions. The selector "selects" a rule from a set of rule distributions (i.e. the components of the mixture), and we call this set the *rulebook*. The rulebook is jointly learned with the rest of CMR and can be inspected and verified at every stage of the learning. Architecturally, the selection is akin to an attention mechanism over a differentiable memory [12].

To this end, let $s \in [1, n_R] \subset \mathbb{N}$ be the indicator of the selected component of the mixture, then:

$$p(r|x) = \sum_s p(r|s)p(s|x) \tag{2}$$

---

[1]In case of encoders that provide additional embeddings other than concept logits, they are modelled as Dirac delta distributions, thus not modelling any uncertainty over them.

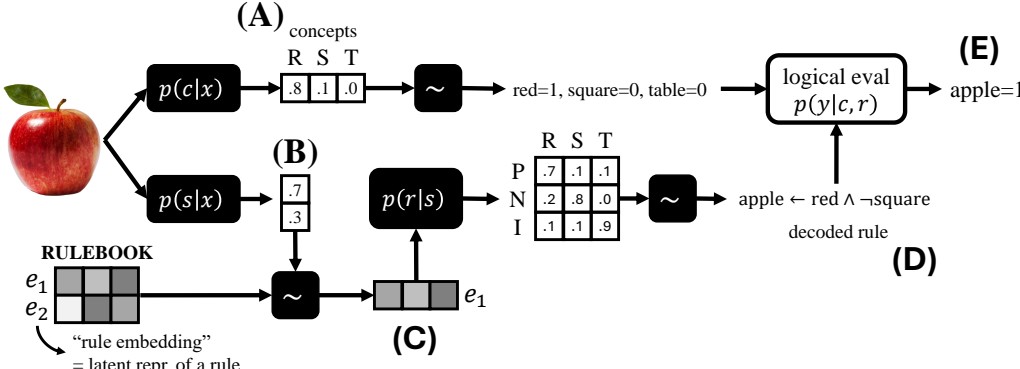

Figure 2: Example prediction of CMR with a rulebook of two rules and three concepts (i.e. $red\ (R)$, $square\ (S)$, $table\ (T)$). In this figure, we sample ($\sim$) for clarity, but in practice, we compute every probability exactly. Every black box is implemented by a neural network, while the white box is a pure symbolic logic evaluation. **(A)** The image is mapped to a concept prediction. **(B)** The image is mapped by the component selector to a distribution over rules. **(C)** This distribution is used to select a rule embedding from the encoded rulebook. **(D)** The rule embedding is decoded into a logic rule by assigning to each of the concepts its role in the rule, i.e. whether it is positive (P), negative (N), or irrelevant (I). Finally, **(E)** the rule is evaluated on the concept prediction to provide the task prediction on the task $apple$.

Here, $p(s|x)$ is the categorical distribution defining the mixing weights of the mixture. It is parameterized by a neural network $\phi^{(s)} : X \to \mathbb{R}^{n_R}$, outputting one logit for each of the $n_R$ components. Each $p(r|s)$ is a distribution over all possible rules and is modelled as a product of $n_C$ categorical distributions, i.e. $p(r|s) = \prod_{i=1}^{n_C} p(r_i|s)$. We assign to each component $s = j$ a *rule embedding* $e_j \in \mathbb{R}^q$, and each categorical $r_i$ is then parameterized by a neural network $\phi_i^{(r)} : \mathbb{R}^q \to \mathbb{R}^3$, mapping the rule embedding to the logits of the categorical component. Intuitively, for each concept $c_i$, the corresponding categorical distribution $p(r_i|s = j)$ *decodes* the embedding $e_j$ to the three possible roles $r_i \in \{P, N, I\}$ of concept $c_i$ in rule $r$. This way, each embedding in the rulebook is the latent representation of a logic rule.[2] Lastly, we define the set $E$ of all rule embeddings, i.e. $E = \{e_j\}_{j \in [1, n_R]}$, as the encoded rulebook.

### 3.1.2 Task predictor

The CMR task predictor $p(y|r, c)$ provides the final $y$ prediction given concept predictions $c$ and the selected rule $r$. We model the task predictor as a degenerate deterministic distribution, providing the entire probability mass to the unique value $y$ which corresponds to the logical evaluation of the rule $r$ on the concept predictions $c$. In particular, let $r_i \in \{P, N, I\}$ be the role of the $i$-th concept in rule $r$. Then, the symbolic task prediction $y$ obtained by evaluating rule $r$ on concept predictions $c$ is:

$$y \leftarrow \bigwedge_{i=1}^{n_C} (r_i = I) \vee (((r_i = P) \Rightarrow c_i) \wedge ((r_i = N) \Rightarrow \neg c_i))) \tag{3}$$

Here, the $y$ prediction is equivalent to a conjunction of $n_C$ different conjuncts, one for each concept. If a concept $i$ is irrelevant according to the selected rule $r$ (i.e. $r_i = I$), the corresponding conjunct is ignored. If $r_i = P$, then the conjunct is True if the corresponding concept is True. Otherwise, i.e. if $r_i = N$, the conjunct is True if the corresponding concept is False.

A graphical representation of the model is shown in Figure 2.

---

[2]Actually, each embedding is the latent representation of a *distribution* over all possible rules, factorized per concept. However, this distribution is typically quite crisp after training (see also Appendix B). After training, we convert each distribution into the most likely rule by taking the argmax among the roles of the concepts (see Section 5). This way, each embedding in the encoded rulebook corresponds with a rule, and the decoded rulebook is a set of rules.

# 4 Expressivity, interpretability and verification

In this section, we will discuss the proposed model along three different directions: *expressivity*, *interpretability* and *verification*.

## 4.1 Expressivity

An interesting property is that CMR is as expressive as a neural network binary classifier.

**Theorem 4.1.** *CMR is a universal binary classifier [13] if $n_R \geq 3$.*

*Proof.* Recall that the rule selector is implemented by some neural network $\phi^{(s)} : X \to \mathbb{R}^{n_R}$. Consider the following three rules, easily expressible in CMR as showed on the right of each rule:

$$y \leftarrow True \ (\text{i.e.} \ \forall i : r_i = I), \quad y \leftarrow \bigwedge_{i=1}^{n_C} c_i \ (\text{i.e.} \ \forall i : r_i = P), \quad y \leftarrow \bigwedge_{i=1}^{n_C} \neg c_i \ (\text{i.e.} \ \forall i : r_i = N)$$

By selecting one of these three rules, the rule selector can always make a desired $y$ prediction, regardless of the concepts $c$. In particular, to predict $y = 1$, the selector can select the first rule. To predict $y = 0$ when at least one concept has probability less than 50% in the concept predictions $c$ (i.e. $\exists i : p(c_i|x) < 0.50$), it can select the second rule. Lastly, to predict $y = 0$ when all concepts have probability of at least 50% in $c$ (i.e. $\forall i : p(c_i|x) \geq 0.50$), it can select the last rule. $\square$

Consequently, CMR can in theory always achieve the same accuracy as a neural network *without concept bottleneck*, irrespective of which concepts are employed in the model. This distinguishes CMR sharply from CBNMs.

## 4.2 Interpretability

CMR's task prediction is the composition of a (neural) rule selector and the symbolic evaluation of the selected rule. Therefore, we can always inspect the whole rulebook to know exactly the global behaviour of the task prediction. In particular, let $s$ be the selected rule, $e_j$ the embedding of the $j$-th rule, and $r_i^{(j)} \in \{P, N, I\}$ the role of the $i$-th concept in the $j$-th rule at decision time, i.e. $r_i^{(j)} = \text{argmax}(\phi_i^{(r)}(e_j))$. Then, CMR's task prediction can be logically defined as the global rule obtained as the disjunction of all decoded rules, each filtered by whether the rule has been selected or not. That is:

$$y \Leftrightarrow \bigvee_{j=1}^{n_R} (s = j) \wedge \left( \bigwedge_{i=1}^{n_C} (r_i^{(j)} = I) \vee (((r_i^{(j)} = P) \Rightarrow c_i) \wedge ((r_i^{(j)} = N) \Rightarrow \neg c_i)) \right) \quad (4)$$

It is clear that if the model learns the three rules in the proof of Theorem 4.1, the selector simply becomes a *proxy* for $y$. It is safe to say that the interpretability of the selector (and consequently of CMR) fully depends on the interpretability of the learned rules.

**Prototype-based rules.** In order to develop interpretable rules, we focus on standard theories in cognitive science [14] by looking at those rules that are *prototypical* of the concept activations on which they are applied to. Prototype-based models are often considered one of the main categories of interpretable models [9] and have been investigated in the context of concept-based models [15]. However, CMR distinguishes itself from prototype-based models in two significant ways. First, in prototype-based CBMs, prototypes are built directly in the input space, such as images [16] or on part of it [17], and are often used to automatically build concepts. However, this approach carries similar issues w.r.t. traditional input-based explanations, like saliency maps (i.e. a prototype made by a strange pattern of pixels can still remain unclear to the user). In contrast, in CMR, prototypes (i.e. rules) are built on top of human-understandable concepts and are therefore interpretable as well. Second, differently from prototype-based models, CMR assigns a logical interpretation to prototypes as conjunctive rules. Unlike prototype networks that assign class labels only based on proximity to a prototype, CMR determines class labels through the symbolic evaluation of prototype rules on concepts. Therefore, prototypes should be representative of concept activations *but also* provide a correct task prediction. This dual role of CMR prototypes adds a constraint: only prototypes of positive instances (i.e. $y = 1$) can serve as effective classification rules. Thus, when rules are selected for negative instances (i.e. $y = 0$), they do not need to be representative of the concept predictions.

**Interventions.** In contrast to existing CBMs, which only allow prediction-time concept interventions, the global interpretability of CMR allows humans to intervene also on the rules that will be exploited for task prediction. These interactions may occur during the training phase to directly shape the model that is being learned, and may come in various forms. A first approach involves the manual inclusion of rules into the rulebook, thereby integrating expert knowledge into the model [18]. The selector mechanism within the model learns to choose among the rules acquired through training and those manually incorporated. As a result, the rules that are being acquired through training can change after adding expert rules. A second approach consists of modifying the learned rules themselves, by altering the role $r_i$ of a concept within a rule. For instance, setting the logit of $P_i$ to 0 ensures that $c_i$ cannot exhibit a positive role in that rule, and setting the logits of both $P_i$ and $N_i$ to 0 ensures irrelevance. This type of intervention could be exploited to remove (or prevent) biases and ensure counterfactual fairness [19].

### 4.3 Verification

One of the main properties of CMR is that, at decision time, it explicitly represents the task prediction as a set of conjunctive rules. Logically, the mixture semantics of the selector can be interpreted as a disjunction, leading to a standard semantics in stochastic logic programming [20, 21]. As the only neural component is in the selection and the concept predictions, task predictions generated using CMR's global formula (cf. Section 4.2) can be automatically verified by using standard tools of formal verification (e.g. model checking), no matter which rule *will* be selected. Being able to verify properties prior to deployment of the model strongly sets CMR apart from existing models, where verification tasks can only be applied at prediction time. In particular, given any propositional logical formula $\alpha$ over the propositional language $\{c_1, c_2, ..., c_{n_C}, y\}$, $\alpha$ can be automatically verified to logically follow from Equation 4. This formula can be converted into a propositional one by (1) evaluating the role expressions (i.e. $r_i^j = \cdot$ becomes *True* or *False*), (2) replacing the selection expressions with a new propositional variable per rule (i.e. $(s = j)$ becomes $s_j$), and (3) adding mutual exclusivity constraints for the different $s_j$. For example, for $n_C = n_R = 2, r_1^1 = P, r_2^1 = I, r_1^2 = P, r_2^2 = N$, we get:

$$(s_1 \oplus s_2) \wedge (y \Leftrightarrow (s_1 \wedge c_1) \vee (s_2 \wedge c_1 \wedge \neg c_2)) \tag{5}$$

with $\oplus$ the xor connective. In logical terms, if the formula $\alpha$ is entailed by such a formula, it means that $\alpha$ must be true every time Equation 4 is used for prediction.

## 5 Learning and inference

### 5.1 Learning problem

Learning in CMR follows the standard objective in CBM literature, where the likelihood of the concepts and task observations in the data is maximized. Formally, let $\Omega$ be the set of parameters of the probability distributions, such that CMR's probabilistic graphical model is globally parameterized by $\Omega$, i.e. $p(y, r, c|x; \Omega)$. Let $D = \{(\hat{x}, \hat{c}, \hat{y})\}$ be a concept-based dataset of i.i.d. triples (input, concepts, task). Then, learning is the optimization problem:

$$\max_{\Omega} \sum_{(\hat{x}, \hat{c}, \hat{y}) \in D} \log p(\hat{y}, \hat{c}|\hat{x}; \Omega) \tag{6}$$

Due to the factorization of the mixture model in the rule selection, CMR has a tractable likelihood computation. In particular, the following result holds.

**Theorem 5.1** (Log-likelihood). *The maximum likelihood simplifies to the following $O(n_C \cdot n_R)$ objective:*

$$\max_{\Omega} \sum_{(\hat{x}, \hat{c}, \hat{y}) \in D} \left( \sum_{i=1}^{n_C} \log p(c_i = \hat{c}_i|\hat{x}) \right) + \left( \log \sum_{\hat{s}=1}^{n_R} p(s = \hat{s}|\hat{x}) \, p(y = \hat{y}|\hat{c}, \hat{s}) \right) \tag{7}$$

*with:*

$$p(y = 1|c, s) = \prod_{i=1}^{n_C} \left( p(I_i|s) + p(P_i|s) \, \mathbb{1}[c_i = 1] + p(N_i|s) \, \mathbb{1}[c_i = 0] \right)$$

*where $\mathbb{1}[\cdot]$ is an indicator function of the condition within brackets.*

*Proof.* See Appendix A □

The maximum likelihood approach only focuses on the prediction accuracy of the model. However, as discussed in Section 4.2, we look for the set of learned rules $r$ to represent prototypes of concept predictions, as in prototype-based learning [15]. To drive the learning of representative positive prototypes when we observe a positive value for the task, i.e. when $y = 1$, we add a regularization term to the objective. Intuitively, every time a rule is selected for a given input instance $x$ with task label $y = 1$, we want the rule to be as close as possible to the observed concept prediction. At the same time, since the number of rules is limited and the possible concept activations are combinatorial, the same rule is expected to be selected for different concept activations. When this happens, we will favour rules that assign an irrelevant role to the inconsistent concepts in the activations. The regularized objective is:

$$\max_{\Omega} \sum_{(\hat{x},\hat{c},\hat{y}) \in D} \left( \sum_{i=1}^{n_C} \log p(c_i = \hat{c}_i | \hat{x}) \right) + \left( \log \sum_{\hat{s}=1}^{n_R} p(s = \hat{s} | \hat{x}) \, p(y = \hat{y} | \hat{c}, \hat{s}) \underbrace{p_{reg}(r = \hat{c} | \hat{s})^{\hat{y}}}_{\text{Regularization Term}} \right) \quad (8)$$

and:

$$p_{reg}(r = \hat{c} | s) = \prod_{i=1}^{n_C} (0.5 \, p(r_i = I | s) + p(r_i = P | s) \, \mathbb{1}[\hat{c}_i = 1] + p(r_i = N | s) \, \mathbb{1}[\hat{c}_i = 0])$$

This term favours the selected rule $r$ to reconstruct the observed $\hat{c}$ as much as possible. When such reconstruction is not feasible due to the limited capacity of the rulebook, the term will favour irrelevant roles for concepts. In this way, we will develop rules that have relevant terms (i.e. $r_i \in \{P, N\}$) only if they are representative of all the instances in which the rule is selected. Appendix B contains an investigation of the influence of this regularization on the optima of the loss.

## 5.2 Inference

After training, we replace each role distribution $p(r_i | s = j)$ for each concept $i$ and rule embedding $j$ with the most likely role. This ensures that each embedding corresponds to a single logic rule rather than a distribution over all possible rules. Moreover, at decision-time, the concepts are unobserved, leading to the following likelihood computation:

$$p(y = 1 | x) = \sum_{\hat{s}=1}^{n_R} p(s = \hat{s} | x) \prod_{i=1}^{n_C} (\mathbb{1}[r_i = I] + \mathbb{1}[r_i = P] \, p(c_i | x) + \mathbb{1}[r_i = N] \, p(\neg c_i | x)) \quad (9)$$

with $r_i = \text{argmax}_{\hat{r}_i \in \{P,N,I\}} p(r_i = \hat{r}_i | s = \hat{s})$.

# 6 Experiments

Our experiments aim to answer the following research questions:

(1) **Generalization:** Does CMR attain similar task and concept accuracy as existing CBMs and black boxes? Does CMR generalize well when the concept set is incomplete[3]?

(2) **Explainability and Intervenability:** Can CMR recover ground truth rules? Can CMR learn meaningful rules when the concept set is incomplete? Are concept interventions and rule interventions effective in CMR?

(3) **Verifiability:** Can CMR allow for post-training verification regarding its behaviour?

## 6.1 Experimental setting

This section describes essential information about experiments. We provide further details in Appendix C.

---

[3]Incomplete concept sets do not contain all the information present in the input that is useful for task prediction. Models with a concept bottleneck cannot achieve black-box accuracy with them.

Table 1: Task accuracy on all datasets. The best and second best for CBMs are shown in bold (black and purple, respectively).

| | MNIST+ | MNIST+$^*$ | C-MNIST | CELEBA | CUB | CEBAB |
|---|---|---|---|---|---|---|
| CBM+linear | $0.00_{\pm 0.00}$ | $0.00_{\pm 0.00}$ | $99.07_{\pm 0.31}$ | $49.02_{\pm 0.20}$ | $50.60_{\pm 0.69}$ | $45.15_{\pm 29.93}$ |
| CBM+MLP | $\mathbf{97.41}_{\pm 0.55}$ | $72.51_{\pm 2.42}$ | $99.42_{\pm 0.11}$ | $50.29_{\pm 0.60}$ | $55.83_{\pm 0.33}$ | $83.70_{\pm 0.30}$ |
| CBM+DT | $96.73_{\pm 0.39}$ | $77.63_{\pm 0.44}$ | $\mathbf{99.44}_{\pm 0.03}$ | $49.60_{\pm 0.20}$ | $51.83_{\pm 0.48}$ | $83.15_{\pm 0.15}$ |
| CBM+XG | $96.73_{\pm 0.39}$ | $76.54_{\pm 0.54}$ | $\mathbf{99.44}_{\pm 0.03}$ | $50.39_{\pm 0.24}$ | $\mathbf{62.97}_{\pm 0.96}$ | $83.80_{\pm 0.01}$ |
| CEM | $92.44_{\pm 0.26}$ | $92.94_{\pm 1.15}$ | $99.32_{\pm 0.11}$ | $\mathbf{65.44}_{\pm 0.25}$ | $57.03_{\pm 0.80}$ | $83.30_{\pm 1.59}$ |
| DCR | $90.70_{\pm 1.21}$ | $92.24_{\pm 1.37}$ | $98.99_{\pm 0.08}$ | $35.65_{\pm 1.53}$ | $50.00_{\pm 0.00}$ | $67.30_{\pm 0.93}$ |
| Black box | $83.26_{\pm 8.71}$ | $83.26_{\pm 8.71}$ | $99.19_{\pm 0.11}$ | $65.33_{\pm 0.60}$ | $64.07_{\pm 0.33}$ | $88.67_{\pm 0.19}$ |
| **CMR (ours)** | $\mathbf{97.25}_{\pm 0.24}$ | $\mathbf{94.65}_{\pm 1.99}$ | $99.12_{\pm 0.04}$ | $63.17_{\pm 1.13}$ | $60.07_{\pm 1.70}$ | $\mathbf{85.14}_{\pm 0.43}$ |

**Data & task setup.** We base our experiments on four different datasets commonly used to evaluate CBMs: MNIST+ [22], where the task is to predict the sum of two digits; C-MNIST, where we adapted MNIST to the task of predicting whether a coloured digit is even or odd; MNIST+$^*$, where we removed the concepts for the digits 0 and 1 from the concept set; CelebA [23], a large-scale face attributes dataset with more than 200K celebrity images, each with 40 concept annotations[4]; CUB [24], where the task is to predict bird species based on bird characteristics; and CEBaB [25], a text-based task where reviews are classified as positive or negative based on different criteria (e.g. food, ambience, service, etc). These datasets range across different concept set quality, i.e. complete (MNIST+, C-MNIST, CUB) vs incomplete (CelebA, MNIST+$^*$), and different complexities of the concept prediction task, i.e. easy (MNIST+, MNIST+$^*$, C-MNIST), medium (CEBaB) and hard (CelebA, CUB). All the datasets come with full concept annotations.

**Evaluation.** To measure classification performance on tasks and concepts, we compute subset accuracy and regular accuracy, respectively. For CUB, we instead compute the Area Under the Receiver Operating Characteristic Curve [26] for the tasks due to the large class imbalance. All metrics are reported using the mean and the standard error of the mean over three different runs with different initializations.

**Baselines.** In our experiments, we compare CMR with existing CBM architectures. We consider Concept Bottleneck Models with different task predictors: linear, multi-layer (MLP), decision-tree (DT) and XGBoost (XG). Moreover, we add two state-of-the-art CBMs: Concept Embedding Models (CEM) [27] and Deep Concept Reasoner (DCR) [11]. We employ hard concepts in CMR and our competitors, avoiding the problem of input distribution leakage that can affect task accuracy [28, 29] (see Appendix C for additional details). Finally, we include a deep neural network without concepts to measure the effect of an interpretable architecture on generalization performance.

We provide an additional experiment serving as an ablation study on CMR's joint rule learning in Appendix C.3.3.

## 6.2 Key findings & results

### 6.2.1 Generalization

**CMR's high degree of interpretability does not harm accuracy, which is similar to or better than competitors'.** In Table 1, we compare CMR with its competitors regarding task accuracy. On all data sets, CMR achieves an accuracy close to black-box accuracy, either beating its concept-based competitors or obtaining similar results. In Table 5 of Appendix C, we show that CMR's training does not harm concept accuracy, which is similar to its competitors. Moreover, we provide an experiment showing that CMR's accuracy is robust to the chosen number of rules in Appendix C.3.4.

**CMR obtains accuracy competitive with black boxes even on incomplete concept sets.** We evaluate the performance of CMR on settings with increasingly more incomplete concept sets. Firstly, as shown in Table 1, in MNIST+$^*$, CMR still obtains task accuracy close to the complete setting, beating its competitors which suffer from a concept bottleneck. Secondly, we run an experiment on

---

[4]We remove the concepts $Wavy\_Hair$, $Black\_Hair$ and $Male$ from the concept set and instead use them as tasks.

CelebA where we gradually decrease the number of concepts in the concept set. Figure 3 shows the achieved task accuracies for CMR and the competitors. CMR's accuracy remains high no matter the size of the concept set, while the performance of the competitors with a bottleneck (i.e. all except CEM) strongly degrades.

### 6.2.2 Explanations and intervention

**CMR discovers ground truth rules.** We quantitatively evaluate the correctness of the rules CMR learns on MNIST+ and C-MNIST. In the former, the ground truth rules have no irrelevant concepts; in the latter, they do. In all runs of these experiments, CMR finds all correct ground truth rules. In C-MNIST, CMR correctly learns that the concepts related to colour are irrelevant for classifying the digit as even or odd (see Table 2).

**CMR discovers meaningful rules in the absence of ground truth.** While the other datasets do not provide ground truth rules, a qualitative inspection shows that they are still meaningful. Table 2 shows two examples for CEBaB, and additional rules can be found in Appendix C.

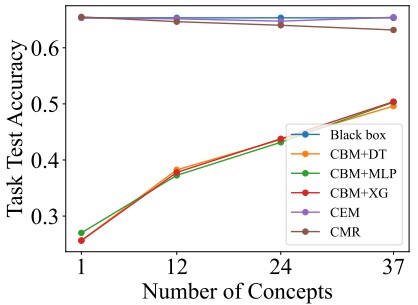

Figure 3: Task accuracy on CelebA with varying numbers of employed concepts.

Table 2: Selection of learned rules. For brevity in C-MNIST, negated concepts are not shown and irrelevant concepts are shown between parentheses. We abbreviate *good* to *g*, *bad* to *b* and *unknown* to *u*.

| | |
|---|---|
| C-MNIST | $y_{even} \leftarrow 0 \wedge (red) \wedge (green)$ 
 $y_{even} \leftarrow 2 \wedge (red) \wedge (green)$ 
 $y_{odd} \leftarrow 3 \wedge (red) \wedge (green)$ |
| CEBaB | $y_{neg} \leftarrow \neg food_g \wedge \neg amb_g \wedge \neg noise_g$ 
 $y_{pos} \leftarrow \neg food_b \wedge \neg amb_b \wedge noise_u \wedge$ 
 $\neg noise_b \wedge \neg noise_g$ |

**Rule interventions during training allow human experts to improve the learned rules.** We show this by choosing a rulebook size for MNIST+ that is too small to learn all ground truth rules. Consequently, CMR learns rules that differ from the ground truth rules. After manually adding rules to the pool in the middle of training, CMR (1) learns to select these new rules for training examples for which they make a correct task prediction, and (2) improves its previously learned rules by eventually converging to the ground truth rules. This is the case for all runs. Table 3 gives some examples of how manually adding rules affects the learned rules. In Appendix C.3.3, we provide an additional experiment with rule interventions, where we add rules extracted from a rule learner. Additionally, as concept interventions are considered a core advantage of CBMs, we show in Appendix C that CMR is equally responsive as competitors, consistently improving its accuracy after concept interventions.

Table 3: Selection of rule interventions and their effect on learned rules. For brevity, negated concepts are not shown, and irrelevant concepts are shown between parentheses.

| Learned rule before intervention | Added rule | Learned rule after intervention |
|---|---|---|
| $y_8 \leftarrow (c_{0,3}) \wedge (c_{1,5}) \wedge (c_{0,4}) \wedge (c_{1,4})$ | $y_8 \leftarrow c_{0,3} \wedge c_{1,5}$ | $y_8 \leftarrow c_{0,4} \wedge c_{1,4}$ |
| $y_9 \leftarrow (c_{0,8}) \wedge (c_{1,1}) \wedge (c_{0,1}) \wedge (c_{1,8})$ | $y_9 \leftarrow c_{0,8} \wedge c_{1,1}$ | $y_9 \leftarrow c_{0,1} \wedge c_{1,8}$ |
| $y_9 \leftarrow (c_{0,0}) \wedge (c_{1,9}) \wedge (c_{0,2}) \wedge (c_{1,7})$ | $y_9 \leftarrow c_{0,0} \wedge c_{1,9}$ | $y_9 \leftarrow c_{0,2} \wedge c_{1,7}$ |

### 6.3 Verification

**CMR allows verification of desired global properties.** In this task, we automatically verify semantic consistency properties for MNIST+ and CelebA whether CMR's task prediction satisfies some properties of interest. For verification, we exploited a naive model checker that verifies whether the property holds for all concept assignments where the theory holds. When this is not feasible, state-of-the-art model formal verification tools can be exploited, as both the task prediction and the property are simply two propositional formulas. For MNIST+, we can verify that, for each task $y$, CMR never uses more than one positive concept (i.e. digit) per image. This can be done by

verifying one formula per concept $j$ of digit $k$: $\forall y, i \neq j : y \wedge c_{k,j} \Rightarrow \neg c_{k,i}$. This is also easily verifiable by simply inspecting the rules in Appendix C. Moreover, in CelebA, we can easily verify that $Bald \Rightarrow \neg\,Wavy\_Hair$ with the learned rulebook for $n_C = 12$ (see Table 10 in Appendix C), as $\neg Bald$ is a conjunct in each rule that does not trivially evaluate to False.

# 7   Related works

In recent years, XAI techniques have been criticized for their vulnerability to data modifications [30, 31], insensitivity to reparametrizations, [32], and lacking meaningful interpretations for non-expert users [33]. To address these issues, Concept-based methods [34, 35, 5, 10] have emerged, offering explanations in terms of human-interpretable features, a.k.a. concepts. Concept Bottleneck Models [4] go a step further by directly integrating these concepts as explicit intermediate network representations. Concept Embeddings Models (CEMs) [7, 8, 11] close the accuracy gap with black-box models through vectorial concept representations. However, they still harm the interpretability, as it is unclear what information is contained in the embeddings. In contrast, CMR closes the accuracy gap by using a neural rule selector coupled with learned symbolic logic rules. As a result, CMR's task prediction is transparent, allowing experts to see *how* concepts are being used for task prediction, and allowing intervention and automatic verification of desired properties. To the best of our knowledge, there is only one other attempt at analysing CBMs' task prediction in terms of logical formulae, namely DCR [11]. For a given example, DCR *predicts* and subsequently evaluates a (fuzzy) logic rule. As rules are predicted on a per-example basis, the global behaviour of DCR cannot be inspected, rendering interaction (e.g. adding expert rules) and verification impossible. In contrast, CMR *learns* (probabilistic) logic rules in a memory, allowing for inspection, interaction and verification.

The use of logic rules by CMR for interpretability purposes aligns it closely with the field of neurosymbolic AI [36, 37]. Here, logic rules [38, 39, 18] or logic programs [40, 22, 21] are used in combination with neural networks through the use of neural predicates [22]. Concepts in CMR are akin to a propositional version of neural predicates. However, in CMR, the set of rules is learned instead of given by the human and direct concept supervision is used for human alignment. While neurosymbolic rule learning methods have been developed, many are constrained by specific assumptions about the nature of the task, limiting their usability to particular datasets or environments (e.g. requiring multitask scenarios [41] or specific datasets like MNIST [42]). Additionally, some approaches, unlike ours, explore the rule space in a discrete manner [43], which is computationally expensive. Furthermore, they do not provide expressivity results, while we show that CMR is a universal binary classifier.

Finally, the relationships with prototype-based models have already been discussed in Section 4.2.

# 8   Conclusion

We propose CMR, a novel Concept-Based Model that offers a human-understandable and provably-verifiable task prediction process. CMR integrates a neural selection mechanism over a memory of learnable logic rules, followed by a symbolic evaluation of the selected rule. This approach enables global interpretability and verification of task prediction properties. Our results show that (1) CMR achieves near-black-box accuracy, (2) discovers meaningful rules, and (3) facilitates strong interaction with human experts through rule interventions. The development of CMR can have significant societal impact by enhancing transparency, verifiability, and human-AI interaction, thereby fostering trust and reliability in critical decision-making processes.

**Limitations and future works.** CMRs are still fundamental models and several limitations need to be explored further in future works. In particular, CMRs focus on positive-only explanations, while negative-reasoning explanations have not been explored yet. Moreover, the same selection mechanism can be tested in non-logic, globally interpretable settings (like linear models). Finally, the verification capabilities of CMR will be tested on more realistic, safety critical domains, where the model can be verified against safety specifications.

## Acknowledgments and Disclosure of Funding

DD is a fellow of the Research Foundation-Flanders (FWO-Vlaanderen, 1185125N). This research has also received funding from the KU Leuven Research Fund (STG/22/021, CELSA/24/008) and from the Flemish Government under the "Onderzoeksprogramma Artificiële Intelligentie (AI) Vlaanderen" programme. FG has been supported by the Partnership Extended PE00000013 - "FAIR - Future Artificial Intelligence Research" - Spoke 1 "Human-centered AI". PB acknowledges support from SNSF project TRUST-ME (No. 205121L_214991). MD was supported by TAILOR and by HumanE-AI-Net projects funded by EU Horizon 2020 research and innovation programme under GA No 952215 and No 952026, respectively. This study has received funding from the European Union's EU Framework Program for Research and Innovation Horizon under the Grant Agreement No 101073307 (MSCA-DN LeMuR).

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

# A Maximum likelihood derivation

We show that the maximum likelihood problem in Equation 6 simplifies to:

$$\log p(\hat{y}, \hat{c}|\hat{x}) = \left( \sum_{i=1}^{n_C} \log p(c_i = \hat{c}_i|\hat{x}) \right) + \left( \log \sum_{\hat{s}=1}^{n_R} p(s = \hat{s}|\hat{x})\, p(y = \hat{y}|\hat{c}, \hat{s}) \right)$$

with:

$$p(y|c, s) = \prod_{i=1}^{n_C} \left( p(I_i|s) + p(P_i|s)\, \mathbb{1}[c_i = 1] + p(N_i|s)\, \mathbb{1}[c_i = 0] \right)$$

*Proof.* Let $n := n_C$. First, we express the likelihood as the marginalization of the distribution over the unobserved variables

$$p(y, c|x) = p(c|x) \sum_s p(s|x) \sum_{r_1} ... \sum_{r_n} p(r_1, ..., r_n|s) p(y|r_1, ..., r_n, c_1, ..., c_n)$$

Due to the independence of the individual components of the rule distribution:

$$p(r_1, ..., r_n|s) = \prod_{i=1}^{n} p(r_i|s)$$

The logical evaluation of a rule can be expressed in terms of indicator functions over the different variables involved.

$$p(y|r_1, ..., r_n, c_1, ..., c_n) = \prod_{i=1}^{n} \mathbb{1}[r_i = I] + \mathbb{1}[r_i = P]\mathbb{1}[c_i = 1] + \mathbb{1}[r_i = n]\mathbb{1}[c_i = 0]$$

Let us define $f(r_i, c_i) := \mathbb{1}[r_i = I] + \mathbb{1}[r_i = P]\mathbb{1}[c_i = 1] + \mathbb{1}[r_i = n]\mathbb{1}[c_i = 0]$.

Then, the likelihood becomes:

$$
\begin{aligned}
p(y, c|x) &= p(c|x) \sum_s p(s|x) \sum_{r_1} ... \sum_{r_n} \prod_{i=1}^{n} p(r_i|s)\, f(r_i, c_i) \\
&= p(c|x) \sum_s p(s|x) \left( \sum_{r_1} p(r_1|s)\, f(r_1, c_1) \right) (...) \left( \sum_{r_n} p(r_n|s)\, f(r_n, c_n) \right) \\
&= p(c|x) \sum_s p(s|x) \prod_{i=1}^{n} \sum_{r_i} p(r_i|s)\, f(r_i, c_i) \\
&= p(c|x) \sum_s p(s|x) \prod_{i=1}^{n} \left( p(I_i|s)\, f(I_i, c_i) + p(P_i|s)\, f(P_i, c_i) + p(N_i|s)\, f(N_i, c_i) \right) \\
&= p(c|x) \sum_s p(s|x) \prod_{i=1}^{n} \left( p(I_i|s) + p(P_i|s)\, \mathbb{1}[c_i = 1] + p(N_i|s)\, \mathbb{1}[c_i = 0] \right) \\
&= p(c|x) \sum_s p(s|x)\, p(y|c, s)
\end{aligned}
$$

Using $p(c|x) = \prod_{i=1}^{n_C} p(c_i|x)$, and applying the logarithm, we find:

$$\log p(y, c|x) = \left( \sum_{i=1}^{n_C} \log p(c_i|x) \right) + \left( \log \sum_{s=1}^{n_R} p(s|x)\, p(y|c, s) \right)$$

□

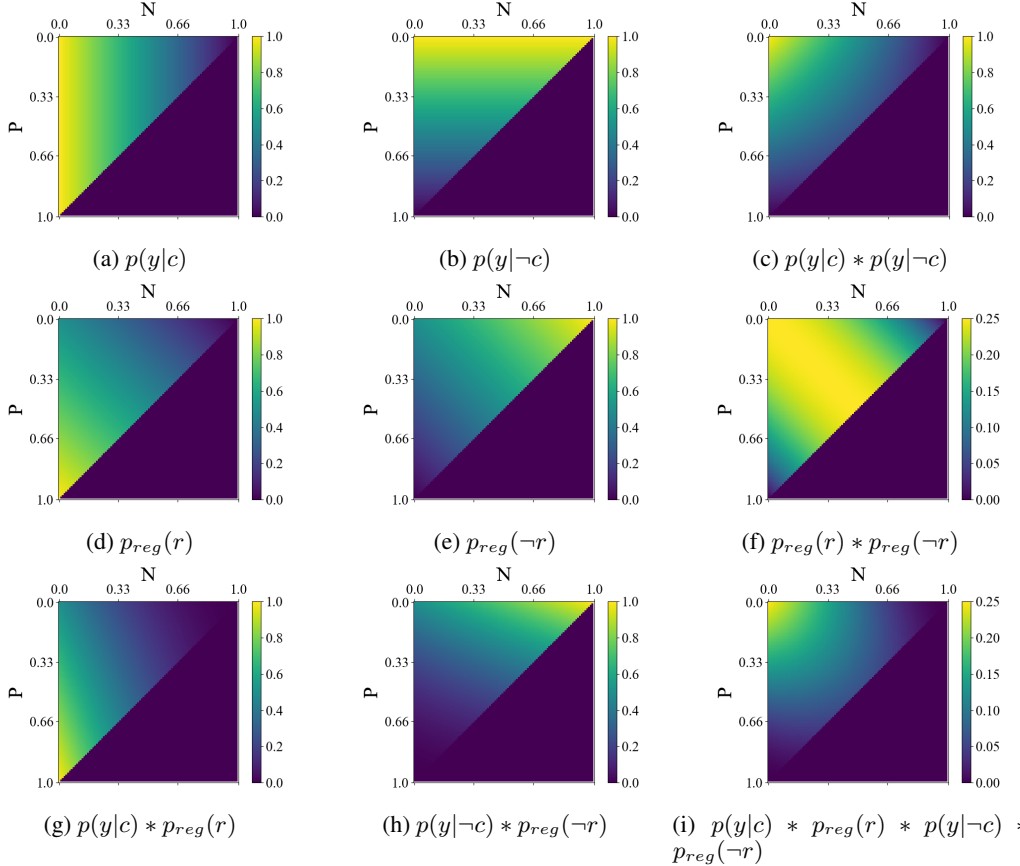

Figure 4: Likelihoods to be maximized when $y = 1$ w.r.t. the role $r$ of a single concept in a selected rule, for different situations. As $P + N + I = 1$, irrelevance is the coordinate $(0, 0)$. Likelihoods that cannot be achieved (i.e. when $P + N > 1$) are put to 0. In the first column, the concept label is 1. In the second column, it is 0. In the third column, two examples with opposite labels select the rule.

## B  Implementation and optimization details

**Selector re-initialization**    To promote exploration, we re-initialize the parameters of $p(s|x)$ multiple times during training, making it easier to escape local optima. The re-initialization frequency is a hyperparameter that differs between experiments, see Appendix C.

**Effect of the regularization term**    Figure 4 shows the probabilities to be maximized when the label $y = 1$, for a selected rule and a single concept, with respect to different roles $r$ for that concept. Figures 4a, 4b and 4c show the probabilities to be maximized without the regularization. Figures 4d, 4e and 4f show the regularization probabilities (remember, we only have these if $y = 1$). Figures 4g, 4h and 4i show the probabilities when both are present. As mentioned in the main text, when $c$ is True for all examples that select the rule, we want that concept's role to be $P$. However, when optimizing without regularization, it is clear that e.g. $I$ is also an optimum (Figure 4a). Because the regularization only has an optimum in $P$ (Figure 4d), adding the regularization results in the correct optimum (Figure 4g). A similar reasoning applies when $c$ is False for all examples that select the rule; consider Figures 4b, 4e and 4h. It should be noted that the regularization alone is insufficient to get the correct optima; indeed, when examples with *different* $c$ select the rule, the regularization term has many optima (Figure 4f), of which only $I$ is desired (as explained in the main text). As the original loss only has an optimum in $I$ in this case (Figure 4c), the only optimum that remains is the correct one (Figure 4i). **In summary:** the loss function we use has the desired optima (last row in Figure 4), dropping the regularization (first row in Figure 4) or only keeping the regularization (middle row in Figure 4) both have cases where the optima are undesired.

**Passing predicted concepts**   While probabilistic semantics require passing the ground truth $c$ to the task predictor during training (as these are observed variables), CBMs typically instead pass the concept predictions (opposite of teacher forcing) to make the model more robust to its own errors. For this reason, we also employ this technique in CMR in the experiments with CEBaB, CUB and CelebA. For competitors, we always employ this technique. However, we pass hard concept predictions in both CMR and the competitors, as to avoid leakage (see Appendix C).

**Weight in the loss**   We introduce a weight $\beta$ in the regularized objective of Equation 8 (see below). This adjustment helps to balance the regularization and the task loss. For instance, if the regularization dominates, CMR might learn rules that can only be used to predict $y = 1$ in practice. Additionally, a larger $\beta$ might assist in settings where achieving high accuracy is difficult for some concepts; in such cases, concepts with lower accuracy can be more easily considered irrelevant by choosing a larger $\beta$, as they are unable to reliably contribute to correct task prediction. The updated objective is as follows:

$$
\max_{\Omega} \sum_{(\hat{x},\hat{c},\hat{y}) \in D} \left( \sum_{i=1}^{n_C} \log p(c_i = \hat{c}_i | \hat{x}) \right) + \left( \log \sum_{\hat{s}=1}^{n_R} p(s = \hat{s} | \hat{x}) \, p(y = \hat{y} | \hat{c}, \hat{s})^{\beta} \underbrace{p_{reg}(r = \hat{c} | s)^{\hat{y}}}_{\text{Regularization Term}} \right)
\tag{10}
$$

## C   Experiments

### C.1   Datasets

**MNIST+**   This dataset [22] consists of pairs of images, where each image is an MNIST image of a digit and the task is the sum of the two digits. For these tasks, all concepts are relevant. There is a total of 30,000 training examples and 5,000 test examples.

**MNIST+**$^{*}$   We create this dataset as MNIST+ except that the concepts for digits 0 and 1 are removed from the concept set. This makes the concept set incomplete.

**C-MNIST**   We derive this dataset from MNIST [44], taking the MNIST training (60,000 examples) and test set (10,000 examples), randomly colouring each digit either red or green, and adding these two colours as concepts. There are two tasks: The first is whether the digit is even, and the second is whether it is odd. For these tasks, the concepts related to colour are irrelevant.

**CelebA**   This is a large-scale face attributes dataset with more than 200K celebrity images [23]. Each image has 40 concept annotations. As tasks, we take the concepts $Wavy\_Hair$, $Black\_Hair$, and $Male$, removing them from the concept set.

**CUB**   In this dataset, the task is to predict bird species from images, where the concepts are 112 bird characteristics such as tail colour, wing pattern, etc [24]. CUB originally consists of 200 tasks, of which we take the first 10. Some concepts are strongly imbalanced (some are True in only $\pm 0.5\%$ of examples, others in $40\%$, etc.) and there is a large task imbalance (each task is True in only $\pm 0.5\%$ of examples).

**CEBaB**   This is a text-based dataset that consists of reviews, where the task is to classify them as positive or negative [25]. There are 12 concepts (food, ambience, service, noise, each either unknown, bad or good) and 2 tasks (positive or negative review).

### C.2   Training

**Reproducibility**   We seed all experiments using seeds 1, 2 and 3.

**Soft vs hard concepts**   When departing from pure probabilistic semantics, CBMs allow not only the use of concepts as binary variables, but they allow for concepts to be passed to the task predictor together with their prediction scores, which is called employing *soft concepts* (vs *hard concepts*). While some CBMs use soft concepts as this results in higher task accuracy, the downside of this

is that the use of soft concepts also comes with the introduction of *input distribution leakage*: The concept probabilities can encode much more information than what is related to the concepts, severely harming the interpretability of the model [28, 29]. For this reason, in our experiments, all models use hard concepts, which is realized by thresholding the soft concept predictions at 50%.

**Model input**    For MNIST+, MNIST+$^*$ and C-MNIST, we train directly on the images. For CelebA and CUB, instead of training on the images, we train the models on pretrained ResNet18 embeddings [45]. Specifically, using the torchvision library, we first resize the images to width and height 224 (using bi-linear interpolation), then normalize them per channel with means $(0.485, 0.456, 0.406)$ and standard-deviations $(0.229, 0.224, 0.225)$ (for CelebA only). Finally, we remove the last (classification) layer from the pretrained ResNet18 model, use the resulting model on each image, and flatten the output, resulting in an embedding. For CEBaB, we use a pretrained BERT model [46] to transform the input into embeddings. Specifically, using the transformers library [47], we create a BERT model for sequence classification from the pretrained model 'bert-base-uncased' with 13 labels (1 per concept and 1 representing both tasks). Then, we fine-tune this model for 10 epochs with batch size 2, 500 warm-up steps, weight decay 0.01 and 8 gradient accumulation steps. After training, we use this model to transform each example into an embedding by outputting the last hidden states for that example.

**General training information**    In all experiments, we use the AdamW optimizer. All neural competitors are optimized to maximize the log-likelihood of the data with a weight on the likelihood of the task (1 if not explicitly mentioned below). After training, for each neural model in CelebA, CEBaB, MNIST+ and MNIST+$^*$, we restored the weights that resulted in the lowest validation loss. In CUB, C-MNIST and the MNIST+ rule intervention experiment, we do not use a validation set, instead restoring the weights that resulted in the lowest training loss. In CelebA, we use a validation split of 8:2, a learning rate of 0.001, a batch size of 1000, and we train for 100 epochs. In CEBaB, we use a validation split of 8:2, a learning rate of 0.001, a batch size of 128, and we train for 100 epochs. In CUB, we use a learning rate of 0.001, a batch size of 1280, and we train for 300 epochs. In MNIST+ and MNIST+$^*$, we use a validation split of 9:1. We use a learning rate of 0.0001, a batch size of 512, and we train for 300 epochs. In C-MNIST, we also use a learning rate of 0.0001, a batch size of 512, and we train for 300 epochs.

**General architecture details**    In CMR, we use two hidden layers with ReLU activation to transform the input into a different embedding. We use 3 hidden layers with ReLU activation and an output layer with Sigmoid activation to transform that embedding into concept predictions. The component $p(s|x)$ takes as input that embedding and is implemented by a single hidden layer with ReLU activation and an output layer outputting $n_{tasks} * n_R$ logits, which are reshaped to $(n_{tasks}, n_R)$ and to which a Softmax is applied over the rule dimension. The rulebook is implemented as an embedding module of shape $(n_{tasks} * n_R, \text{rule emb size})$ that is reshaped to $(n_{tasks}, n_R, \text{rule emb size})$. The rule decoder is implemented by a single hidden layer with ReLU activation and an output layer outputting $3 * n_{concepts}$ logits, which are reshaped to $(n_{concepts}, 3)$, after which a Softmax is applied to the last dimension; the result corresponds to $p(r|s)$. At test time, we make $p(r|s)$ deterministic by setting the probability for the most likely role for each concept to 1 and the others to 0 (effectively collapsing each rule distribution to the most likely rule for that distribution). Then, exactly one rule corresponds with each $s$.

The deep neural network is a feed-forward neural network consisting of some hidden layers with ReLU activation and an output layer with Sigmoid activation.

Both CBM+linear and CBM+MLP have a concept predictor that is a feed-forward neural network using ReLU activation for the 3 hidden layers and a Sigmoid activation for the output layer. For CBM+linear, the task predictor is a single linear layer per task with Sigmoid activation. For CBM+MLP, this is a feed-forward neural network using ReLU activation for the 3 hidden layers, and a Sigmoid activation for the output layer.

For CEM, we use 4 layers with ReLU activation followed by a Concept Embedding Module where the concept embeddings have a size of 30. The task predictor is a feed-forward neural network using ReLU activation for the 3 hidden layers and a sigmoid activation for the output layer.

For DCR, we also use 4 layers with ReLU activation to transform the input into an embedding. Then, one layer with ReLU activation and one with Sigmoid activation are used to transform the embedding

into concept predictions. The embedding and the concepts are fed to the task predictor, which is a Concept Reasoning Layer using the product-t norm and a temperature of 10.

For CBM+DT and CBM+XGboost, we train a CBM+MLP and use its concept predictions as training input to the trees.

In the MNIST+ and MNIST+$^*$ experiments, as mentioned earlier, we use as input the two images instead of embeddings. Therefore, in all models, we use a CNN (trained jointly with the rest of the model) to transform the images into an embedding. This CNN consists of a Conv2d layer with 6 output channels and kernel size 5, a MaxPool2d layer with kernel size and stride 2, ReLU activation, a Conv2d layer with 16 output channels and kernel size 5, another MaxPool2d layer with kernel size and stride 2, and another ReLU activation. The result is flattened, and a linear layer is used to output an embedding per image (with size the same as the number of units in the hidden layers of the rest of the models). This results in 2 embeddings (one per image), which are combined using 10 linear layers, each with ReLU activation (except the last one) and again the same number of units (except the first layer, which has 2*number of units). Additionally, the CNN can output concept predictions by applying 3 linear layers with ReLU activation and a linear layer with a Softmax to each image embedding. This results in 10 concept predictions per image, which are concatenated. The final output of the CNN is the tuple of concept predictions and embedding.

In the C-MNIST experiments, the input is a single image. We use a similar CNN architecture as for MNIST+ but with two differences. Firstly, the activation before the concept prediction is a Softmax on only the first 10 concepts (related to the digits), while the activation on the last 2 concepts is Sigmoid. Secondly, after the flattening operation, we use 3 linear layers with ReLU activation and 1 linear layer without activation to transform the embedding into a different one.

**Hyperparameters per experiment**   We use *number of hidden units* and *embedding size* interchangeably. In CelebA, we always use 500 units in each hidden layer, except for CBM+linear and CMR where we use 100 units instead. For the deep neural network, we use 10 hidden layers. CMR uses a rule embedding size of 100, at most 5 rules per task, a $\beta$ of 30, and we reset the selector every 35 epochs. The trained CBM+MLP for CBM+DT and CBM+XGboost has a weight on the task of 0.01.

In CEBaB, we use 300 units in the hidden layers, except for CMR where we use 100 units instead. For the deep neural network, we use 10 hidden layers. We additionally put a weight on the task loss of 0.01. For CMR, we use a rule embedding size of 100, at most 15 rules per task, a $\beta$ of 4, and we reset the selector every 10 epochs. For CBM+DT and CBM+XGboost, the trained CBM+MLP has a weight on the task of 0.01.

In CUB. we use 100 units in the hidden layers, the deep neural network has 2 hidden layers, and we use a weight on the task loss of 0.01 for concept-based competitors. For CMR, we use a $\beta$ of 3, and additionally down-weigh the loss for negative instances with a weight of 0.005 to deal with the class imbalance. We use a rule embedding size of 500, at most 3 rules per task, and we reset the selector every 25 epochs. For CBM+XGBoost, we add a weight of 200 for the positive instances to deal with the class imbalance. For CBM+DT and CBM+XGboost, we do not train a CBM+MLP, instead using CMR's concept predictor.

In MNIST+ and MNIST+$^*$, we use the CNN as described earlier. We always use 500 units in the hidden layers, except for CBM+linear and CMR where we use 100 units. For the deep neural network, we use 10 hidden layers. For CMR, we use a rule embedding size of 1000, at most 20 rules per task, we reset the selector every 40 epochs, and $\beta$ is 0.1. In MNIST+ specifically, instead of passing the CNN's output embedding to the rule selector $p(s|x)$, we pass the CNN's concept predictions, showing that this alternative can also be used effectively when using a complete concept set. For CBM+DT and CBM+XGboost, we train a CBM+MLP with weight on the task of 0.01 and use its concept predictions as input to the trees. For the rule intervention experiment on MNIST+ (where we give CMR a rulebook size that is too small to learn all ground truth rules), we allow it to learn at most 9 rules per task. Here, the input to the selector is the CNN's output embedding. In MNIST+$^*$, we pass the CNN's output embedding to the rule selector.

In C-MNIST, we use the second CNN described above. We always use 500 units in the hidden layers, except for CBM+linear where we use 100 units. For the deep neural network, we use 10 hidden layers. For DCR and CEM, we put a weight on the task loss of 0.01. For CMR, the rule book has at

most 6 rules per task, with rule embeddings of size 1000. We use a $\beta$ of 1, reset the selector every 40 epochs, and additionally put a weight on the concept reconstruction relative to the concept counts. The CBM+MLP we train for CBM+DT and CBM+XGboost has a weight of 0.01 on the task.

**Hyperparameter search**    For CBM+DT, we tune the maximum depth parameter, trying all values between 1 and $n_C$, and report the results with the highest validation accuracy (training accuracy in absence of a validation set). Parameters for the neural models were chosen that result in the highest validation accuracy (training accuracy in absence of a validation set) (for CMR the $\beta$ parameter, rulebook size and rule embedding size were also chosen based on the learned rules). For $\beta$, we searched within the grid $[0.1, 1, 3, 4, 10, 30]$, for the embedding size within the grid $[10, 100, 300, 500, 1000]$, and for the rule embedding size within $[100, 500, 1000]$. For the deep neural network's number of hidden layers, we searched within the grid $[2, 10, 20]$. For unmentioned parameters in the competitors, we used the default values.

**Remaining setup of the rule intervention experiment**    We first train for 300 epochs. Then, we check in an automatic way for rules that differ from the ground truth. A representative example is the rule $y_3 \leftarrow (c_{0,1}) \wedge (c_{1,2}) \wedge (c_{0,3}) \wedge (c_{1,0})$[5], which can be used by the selector for correctly predicting that $1 + 2 = 3$ and $3 + 0 = 3$. After this, we add each missing ground truth rule, *except one*, and let CMR continue training for 100 epochs. Consequently, CMR improves its originally learned rules that differed from ground truth to the ground truth ones and learns to correctly select between the learned and manually added rules. For instance, if CMR originally learned a rule $y_3 \leftarrow (c_{0,1}) \wedge (c_{1,2}) \wedge (c_{0,3}) \wedge (c_{1,0})$, after manually adding $y_3 \leftarrow c_{0,1} \wedge c_{1,2}$ and fine-tuning, the rule improves to $y_3 \leftarrow c_{0,3} \wedge c_{1,0}$, and CMR will no longer select this rule for the examples $3 + 0$, instead selecting the manually added rule.

## C.3    Additional results

### C.3.1    Concept interventions

To measure the effectiveness of concept interventions [4], we report task accuracy before and after replacing the concept predictions with their ground truth. In Table 4, we observe that CMR is responsive to concept interventions: After the interventions, CMR achieves perfect task accuracy, outperforming CEM and DCR. We leave a more extensive investigation of concept interventions for CMR to future work.

Table 4: Task accuracy before and after concept interventions for MNIST+.

|  | BEFORE | AFTER |
|---|---|---|
| CBM+MLP | $97.41_{\pm 0.55}$ | $100.0_{\pm 0.00}$ |
| CEM | $92.44_{\pm 0.26}$ | $94.68_{\pm 0.31}$ |
| DCR | $90.70_{\pm 1.21}$ | $94.11_{\pm 0.83}$ |
| **CMR (ours)** | $97.25_{\pm 0.24}$ | $100.0_{\pm 0.00}$ |

### C.3.2    Concept accuracies

Table 5 gives the concept accuracy of CMR and the competitors on all datasets. CMR's concept accuracy is similar to its competitors across all datasets.

### C.3.3    Rule interventions with a rule learner

This experiment serves as an ablation study on CMR's end-to-end rule learning component. We investigate whether pre-learning rules using an external rule learner and manually integrating them into CMR's memory impacts its accuracy. We employ a decision tree as the rule learner, although

---

[5]In this notation, for brevity, we only show concepts with a role that is positive (of which none are present in this rule example) or irrelevant (between parentheses); concepts with negative roles are not shown and are the remaining ones.

Table 5: Concept accuracies for all datasets.

| | MNIST+ | MNIST+$^*$ | C-MNIST | CELEBA | CUB | CEBAB |
|---|---|---|---|---|---|---|
| CBM+linear | $99.76_{\pm 0.02}$ | $99.73_{\pm 0.02}$ | $99.72_{\pm 0.03}$ | $87.63_{\pm 0.02}$ | $86.24_{\pm 0.05}$ | $92.69_{\pm 0.07}$ |
| CBM+MLP | $99.75_{\pm 0.05}$ | $99.71_{\pm 0.06}$ | $99.80_{\pm 0.02}$ | $87.68_{\pm 0.02}$ | $86.32_{\pm 0.08}$ | $92.67_{\pm 0.07}$ |
| CBM+DT | $99.69_{\pm 0.04}$ | $99.68_{\pm 0.01}$ | $99.81_{\pm 0.01}$ | $87.70_{\pm 0.04}$ | $85.95_{\pm 0.08}$ | $92.83_{\pm 0.02}$ |
| CBM+XGB | $99.69_{\pm 0.04}$ | $99.68_{\pm 0.01}$ | $99.81_{\pm 0.01}$ | $87.70_{\pm 0.02}$ | $85.95_{\pm 0.08}$ | $92.83_{\pm 0.02}$ |
| CEM | $99.26_{\pm 0.04}$ | $99.26_{\pm 0.13}$ | $99.77_{\pm 0.01}$ | $87.57_{\pm 0.04}$ | $85.82_{\pm 0.23}$ | $92.46_{\pm 0.13}$ |
| DCR | $99.21_{\pm 0.09}$ | $99.74_{\pm 0.02}$ | $99.76_{\pm 0.02}$ | $85.35_{\pm 0.26}$ | $85.67_{\pm 0.15}$ | $92.66_{\pm 0.04}$ |
| **CMR (ours)** | $99.74_{\pm 0.02}$ | $99.73_{\pm 0.01}$ | $99.82_{\pm 0.01}$ | $86.80_{\pm 0.22}$ | $85.95_{\pm 0.08}$ | $92.70_{\pm 0.08}$ |

other rule learners could be used as well. We use the CEBaB dataset where we only take the first task and the first 6 concepts.

In Figure 5, we compare CMR's accuracy against 3 alternatives: (1) using a CBM+DT; (2) injecting the rules learned by the CBM+DT for predicting a positive class into CMR while preventing CMR from learning any rules itself; and (3) injecting these same rules but allowing CMR to learn an additional 15 rules.

**CMR's rule learning component enhances accuracy.** When CMR is allowed to learn rules (both CMR (n=15) and DT → CMR (n=15)), its accuracy improves significantly compared to when only the decision tree's rules are used (DT → CMR (n=0)).

**Injecting rules into CMR does not reduce accuracy when CMR is allowed to learn additional rules.** CMR achieves similar levels of accuracy in both cases where it only learns rules (CMR (n=15) and when supplemented with the decision tree's rules (DT → CMR (n=15)).

**CMR's rule selector enhances accuracy beyond the pre-obtained rules.** Even when CMR is restricted from learning any rules (DT → CMR (n=0)), it still performs better than the original CBM+DT (DT) because of the rule selector.

For learning the decision tree (and the corresponding CBM used for predicting the concepts beforehand), we used seed 1.

### C.3.4 Accuracy robustness to the number of rules

In this experiment, we investigate the robustness of CMR's accuracy with respect to the number of rules hyperparameter ($n_R$). We use the CEBaB dataset where we only consider the first task, and we train CMR for different values of $n_R$. Figure 6 shows that similar levels of accuracy are obtained regardless of the chosen $n_R$ value.

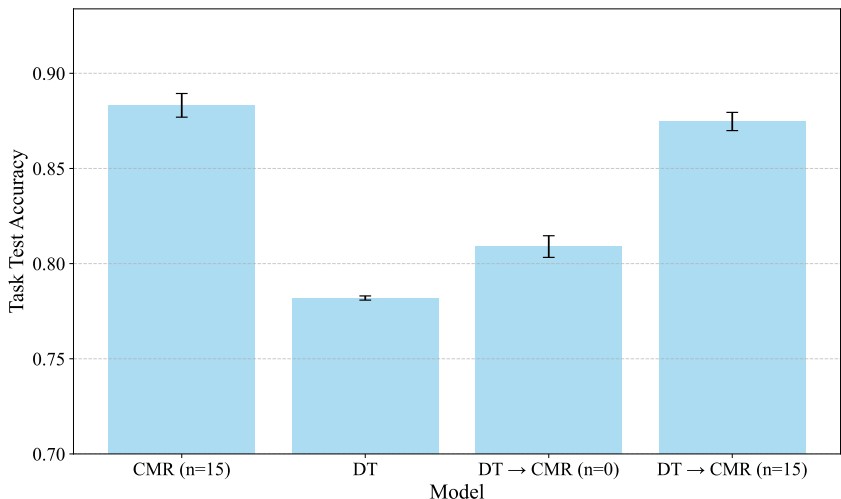

Figure 5: Rule interventions on CEBaB where we predict only the first task and employ 6 concepts. We compare CMR's accuracy with a CBM using a decision tree (DT) and CMR after adding the decision tree's rules to CMR's memory (DT → CMR) without any additional learnable rules (n = 0) and when allowing 15 additional learnable rules (n = 15). The mean and standard deviation is shown over 3 seeds. This means that (1) CMR's end-to-end rule learning allows it to obtain higher accuracy than when purely integrating pre-obtained rules from other rule learners (removing CMR's rule learning component), (2) just integrating pre-obtained rules in CMR (while still allowing more rules to be learned) does not decrease its accuracy, and (3) using CMR with only pre-obtained rules still surpasses the performance of the rules in isolation due to the selector.

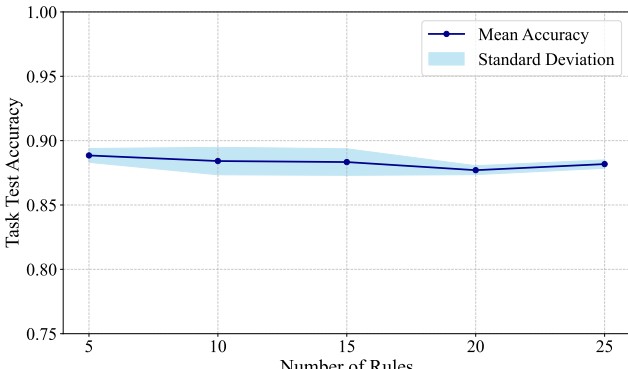

Figure 6: Robustness of CMR's accuracy w.r.t. the number of rules on CEBaB where we predict the first task. The mean and standard deviation is shown over 3 seeds. High accuracy can be obtained regardless.

### C.3.5 Decoded rulebooks

In this section, we provide examples of the decoded rulebooks after training with seed 1 for our experiments. We do not provide decoded rulebooks for MNIST+, as these rules always correspond to the ground truth and are the same over all seeds, i.e. for every task $i$, CMR learns a rule per possible pair of digits that correctly sums to that number $i$. An example of such a rule is $y_3 \leftarrow c_{1,2} \wedge c_{2,1} \wedge D$ where $D$ is a conjunction of $n_C - 2$ conjuncts, where each conjunct is the negation of a different concept. In the tables in this section, we provide the decoded rulebooks for MNIST+$^*$, C-MNIST, CelebA (for all concept subsets, but not the complete rulebook for $n_C = 37$), CUB (not complete either) and CEBaB. Importantly, for brevity, we use a different notation for the rules for C-MNIST, MNIST+$^*$ and CUB: We only show concepts in the rule with as role either positive (as is) or irrelevance (between parentheses); we do not show concepts with negative roles. For the other rulebooks, we use the standard notation.

**C-MNIST**   CMR has learned that the concepts *red* and *green* are irrelevant for predicting *even* or *odd*, learning the (same) ground truth rules for all seeds. We show these rules in Table 6.

**MNIST+**[*]   For brevity, we drop the notation that shows whether the concept is related to the first or second digit. Firstly, consider the tasks $y_0$ and $y_1$. For these tasks, we are essentially missing all the concepts we need for the ground truth rules; we can only predict True by learning the rule that is the negation of all concepts (in the used rule notation, the empty rule), which CMR does. Then, to be able to predict False, it learns some arbitrary rules. Secondly, consider the tasks $y_2$ to $y_{10}$. As we are missing the concepts for digits 0 and 1, we cannot learn all ground truth rules for these tasks. For example, while CMR can (and *does*) still learn the rule $y_4 \leftarrow 2 \wedge 2$, it is impossible to learn a rule like $y_4 \leftarrow 1 \wedge 3$. Instead, CMR learns the rule $y_4 \leftarrow 3$, which can be selected in case it needs to predict that $1 + 3 = 4$. Lastly, consider the tasks $y_{11}$ to $y_{18}$. CMR finds all ground truth rules, as they can be found. We show the learned rules in Figure 7.

**CEBaB**   For predicting $negative$, CMR has learned as one of its rules the empty rule (rule 1), signifying that the other rules (prototypes) it has learned are unable to predict True for all training examples. Similarly, it has learned a rule that always evaluates to False (rules 5, 6 and 7, as food has to be both good and bad). The remaining rules are prototypes. For instance, rule 2 signifies that for some examples if food is bad and service is not good, even though ambiance might not be bad and noise is unknown, the review is negative. Rule 3 signifies that even when food, ambiance and noise are not bad, if service is not good, the review can be negative. We show the learned rules in Table 7.

**CUB**   We show some of the learned rules in Table 8 and some examples that satisfy these rules in Figure 8.

**CelebA**   We show the learned rules in Tables 9, 10 and 11. First consider the rules for $n_C = 1$. For each task, it learns the empty rule, which can always be used to predict True, and two rules that together can always be used to predict False. As mentioned in the main text, these rules cannot be considered meaningful, but since we only have one concept, it makes sense that CMR learns these rules. Consider now the rules for $n_C = 12$. Rules 1, 4 and 7 are rules that always evaluate to False, as $Bald$ and $Blond\_Hair$ cannot be active at the same time. The remaining rules form prototypes. For $n_C = 37$, we also provide a part of the rulebook.

Table 6: Rulebook for C-MNIST.

| |
|---|
| $even \leftarrow 0 \wedge (r) \wedge (g)$ |
| $even \leftarrow 2 \wedge (r) \wedge (g)$ |
| $even \leftarrow 4 \wedge (r) \wedge (g)$ |
| $even \leftarrow 6 \wedge (r) \wedge (g)$ |
| $even \leftarrow 8 \wedge (r) \wedge (g)$ |
| $odd \leftarrow 1 \wedge (r) \wedge (g)$ |
| $odd \leftarrow 3 \wedge (r) \wedge (g)$ |
| $odd \leftarrow 5 \wedge (r) \wedge (g)$ |
| $odd \leftarrow 7 \wedge (r) \wedge (g)$ |
| $odd \leftarrow 9 \wedge (r) \wedge (g)$ |

Figure 7: Rulebook for MNIST+*.

**(a)**

$y_0 \leftarrow$
$y_0 \leftarrow 3 \wedge 5$
$y_0 \leftarrow 6 \wedge 7 \wedge 8 \wedge 9$
$y_0 \leftarrow 7 \wedge 4$
$y_0 \leftarrow 8$

$y_1 \leftarrow$
$y_1 \leftarrow 3$
$y_1 \leftarrow 9 \wedge 9$

$y_2 \leftarrow$
$y_2 \leftarrow 2$

$y_3 \leftarrow 2$
$y_3 \leftarrow 3$

$y_4 \leftarrow 2 \wedge 2$
$y_4 \leftarrow 3$
$y_4 \leftarrow 4$

$y_5 \leftarrow 2 \wedge 3$
$y_5 \leftarrow 3 \wedge 2$
$y_5 \leftarrow 4$
$y_5 \leftarrow 5$

$y_6 \leftarrow 2 \wedge 4$
$y_6 \leftarrow 3 \wedge 3$
$y_6 \leftarrow 4 \wedge 2$
$y_6 \leftarrow 5$
$y_6 \leftarrow 6$

**(b)**

$y_7 \leftarrow 2 \wedge 5$
$y_7 \leftarrow 3 \wedge 4$
$y_7 \leftarrow 4 \wedge 3$
$y_7 \leftarrow 5 \wedge 2$
$y_7 \leftarrow 6$
$y_7 \leftarrow 7$

$y_8 \leftarrow 2 \wedge 6$
$y_8 \leftarrow 3 \wedge 5$
$y_8 \leftarrow 4 \wedge 4$
$y_8 \leftarrow 5 \wedge 3$
$y_8 \leftarrow 6 \wedge 2$
$y_8 \leftarrow 7$
$y_8 \leftarrow 8$

$y_9 \leftarrow 2 \wedge 7$
$y_9 \leftarrow 3 \wedge 6$
$y_9 \leftarrow 4 \wedge 5$
$y_9 \leftarrow 5 \wedge 4$
$y_9 \leftarrow 6 \wedge 3$
$y_9 \leftarrow 7 \wedge 2$
$y_9 \leftarrow 8$
$y_9 \leftarrow 9$

$y_{10} \leftarrow 2 \wedge 8$
$y_{10} \leftarrow 3 \wedge 7$
$y_{10} \leftarrow 4 \wedge 6$

**(c)**

$y_{10} \leftarrow 5 \wedge 5$
$y_{10} \leftarrow 6 \wedge 4$
$y_{10} \leftarrow 7 \wedge 3$
$y_{10} \leftarrow 8 \wedge 2$
$y_{10} \leftarrow 9$

$y_{11} \leftarrow 2 \wedge 9$
$y_{11} \leftarrow 3 \wedge 8$
$y_{11} \leftarrow 4 \wedge 7$
$y_{11} \leftarrow 5 \wedge 6$
$y_{11} \leftarrow 6 \wedge 5$
$y_{11} \leftarrow 7 \wedge 4$
$y_{11} \leftarrow 8 \wedge 3$
$y_{11} \leftarrow 9 \wedge 2$

$y_{12} \leftarrow 3 \wedge 9$
$y_{12} \leftarrow 4 \wedge 8$
$y_{12} \leftarrow 5 \wedge 7$
$y_{12} \leftarrow 6 \wedge 6$
$y_{12} \leftarrow 7 \wedge 5$
$y_{12} \leftarrow 8 \wedge 4$
$y_{12} \leftarrow 9 \wedge 3$

$y_{13} \leftarrow 4 \wedge 9$
$y_{13} \leftarrow 5 \wedge 8$
$y_{13} \leftarrow 6 \wedge 7$

**(d)**

$y_{13} \leftarrow 7 \wedge 6$
$y_{13} \leftarrow 8 \wedge 5$
$y_{13} \leftarrow 9 \wedge 4$

$y_{14} \leftarrow 5 \wedge 9$
$y_{14} \leftarrow 6 \wedge 8$
$y_{14} \leftarrow 7 \wedge 7$
$y_{14} \leftarrow 8 \wedge 6$
$y_{14} \leftarrow 9 \wedge 5$

$y_{15} \leftarrow 6 \wedge 9$
$y_{15} \leftarrow 7 \wedge 8$
$y_{15} \leftarrow 8 \wedge 7$
$y_{15} \leftarrow 9 \wedge 6$

$y_{16} \leftarrow 7 \wedge 9$
$y_{16} \leftarrow 8 \wedge 8$
$y_{16} \leftarrow 9 \wedge 7$

$y_{17} \leftarrow 8 \wedge 9$
$y_{17} \leftarrow 9 \wedge 8$

$y_{18} \leftarrow 9 \wedge 9$

Table 7: Rulebook for CEBAB.

(1) $negative \leftarrow$
(2) $negative \leftarrow \neg food\_unknown \wedge food\_bad \wedge \neg food\_good \wedge \neg ambiance\_bad \wedge \neg service\_good \wedge noise\_unknown \wedge \neg noise\_bad \wedge \neg noise\_good$
(3) $negative \leftarrow \neg food\_bad \wedge \neg ambiance\_bad \wedge \neg service\_good \wedge noise\_unknown \wedge \neg noise\_bad \wedge \neg noise\_good$
(4) $negative \leftarrow \neg food\_unknown \wedge food\_bad \wedge \neg food\_good \wedge \neg ambiance\_good \wedge \neg service\_good \wedge noise\_unknown \wedge \neg noise\_bad \wedge \neg noise\_good$
(5) $negative \leftarrow food\_unknown \wedge food\_bad \wedge food\_good \wedge \neg ambiance\_unknown \wedge ambiance\_bad \wedge \neg ambiance\_good \wedge service\_unknown \wedge service\_bad \wedge service\_good \wedge \neg noise\_unknown \wedge \neg noise\_bad \wedge noise\_good$
(6) $negative \leftarrow food\_unknown \wedge food\_bad \wedge food\_good \wedge \neg ambiance\_unknown \wedge ambiance\_bad \wedge \neg ambiance\_good \wedge service\_unknown \wedge service\_bad \wedge service\_good \wedge \neg noise\_unknown \wedge noise\_bad \wedge noise\_good$
(7) $negative \leftarrow food\_unknown \wedge food\_bad \wedge food\_good \wedge \neg ambiance\_unknown \wedge \neg ambiance\_bad \wedge \neg ambiance\_good \wedge service\_unknown \wedge service\_bad \wedge service\_good \wedge \neg noise\_unknown \wedge \neg noise\_bad \wedge noise\_good$

(8) $positive \leftarrow$
(9) $positive \leftarrow \neg food\_bad \wedge \neg ambiance\_bad \wedge noise\_unknown \wedge \neg noise\_bad \wedge \neg noise\_good$
(10) $positive \leftarrow \neg food\_unknown \wedge food\_bad \wedge \neg food\_good \wedge \neg service\_good \wedge noise\_unknown \wedge \neg noise\_bad \wedge \neg noise\_good$
(11) $positive \leftarrow \neg food\_unknown \wedge food\_bad \wedge \neg food\_good \wedge \neg ambiance\_bad \wedge noise\_unknown \wedge \neg noise\_bad \wedge \neg noise\_good$
(12) $positive \leftarrow food\_unknown \wedge food\_bad \wedge food\_good \wedge \neg ambiance\_unknown \wedge \neg ambiance\_bad \wedge \neg ambiance\_good \wedge service\_unknown \wedge service\_bad \wedge service\_good \wedge noise\_unknown \wedge \neg noise\_bad \wedge noise\_good$

Table 8: Rulebook for CUB (not complete).

(1) $Laysan\_Albatross \leftarrow underparts\_white \wedge breast\_solid \wedge breast\_white \wedge throat\_white \wedge eye\_black \wedge bill\_length\_about\_the\_same\_as\_head \wedge belly\_white \wedge back\_solid \wedge belly\_solid \wedge (bill\_hooked\_seabird) \wedge (wing\_grey) \wedge (wing\_white) \wedge (upperparts\_grey) \wedge (upperparts\_black) \wedge (upperparts\_white) \wedge (back\_brown) \wedge (back\_grey) \wedge (back\_white) \wedge (upper\_tail\_white) \wedge (head\_plain) \wedge (forehead\_black) \wedge (forehead\_white) \wedge (under\_tail\_black) \wedge (under\_tail\_white) \wedge (nape\_white) \wedge (size\_medium\_(9\_16\_in)) \wedge (tail\_solid) \wedge (primary\_grey) \wedge (primary\_black) \wedge (primary\_white) \wedge (bill\_black) \wedge (bill\_buff) \wedge (crown\_black) \wedge (crown\_white) \wedge (wing\_solid)$

(2) $Sooty\_Albatross \leftarrow bill\_hooked\_seabird \wedge wing\_grey \wedge upperparts\_grey \wedge breast\_solid \wedge upper\_tail\_grey \wedge throat\_black \wedge eye\_black \wedge bill\_length\_about\_the\_same\_as\_head \wedge forehead\_black \wedge under\_tail\_grey \wedge under\_tail\_black \wedge size\_medium\_(9\_16\_in) \wedge back\_solid \wedge tail\_solid \wedge belly\_solid \wedge leg\_grey \wedge bill\_black \wedge crown\_black \wedge wing\_solid$

(3) $Brewer\_Blackbird \leftarrow wing\_black \wedge upperparts\_black \wedge breast\_solid \wedge eye\_black \wedge under\_tail\_black \wedge belly\_solid \wedge bill\_black \wedge (bill\_all\_purpose) \wedge (wing\_white) \wedge (underparts\_black) \wedge (back\_black) \wedge (upper\_tail\_black) \wedge (head\_plain) \wedge (breast\_black) \wedge (throat\_black) \wedge (bill\_length\_about\_the\_same\_as\_head) \wedge (bill\_length\_shorter\_than\_head) \wedge (forehead\_black) \wedge (nape\_black) \wedge (belly\_black) \wedge (wing\_rounded\_wings) \wedge (size\_small\_(5\_9\_in)) \wedge (shape\_perching\_like) \wedge (back\_solid) \wedge (tail\_solid) \wedge (primary\_black) \wedge (leg\_black) \wedge (crown\_black) \wedge (wing\_solid)$

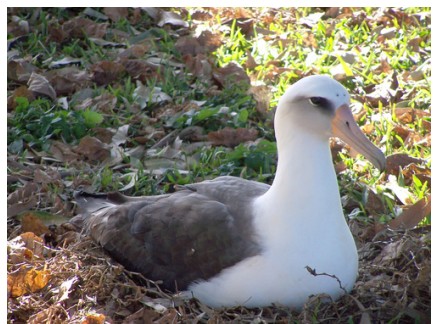

(a)   *Laysan_Albatross ← underparts_white ∧ breast_solid ∧ breast_white ∧ throat_white ∧ eye_black ∧ bill_length_about_the_same_as_head ∧ belly_white ∧ back_solid ∧ belly_solid ∧ (bill_hooked_seabird) ∧ (wing_grey) ∧ (wing_white) ∧ (upperparts_grey) ∧ (upperparts_black) ∧ (upperparts_white) ∧ (...)*

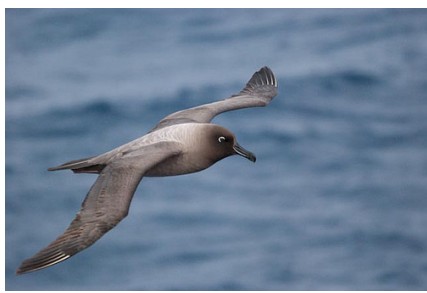

(b) *Sooty_Albatross ← bill_hooked_seabird ∧ wing_grey ∧ upperparts_grey ∧ breast_solid ∧ upper_tail_grey ∧ throat_black ∧ eye_black ∧ bill_length_about_the_same_as_head ∧ forehead_black ∧ under_tail_grey ∧ under_tail_black ∧ size_medium_(9_16_in) ∧ back_solid ∧ tail_solid ∧ belly_solid ∧ leg_grey ∧ bill_black ∧ crown_black ∧ wing_solid*

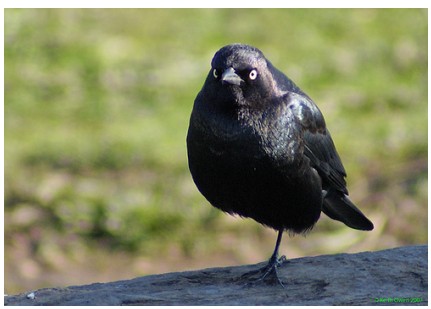

(c) *Brewer_Blackbird ← wing_black ∧ upperparts_black ∧ breast_solid ∧ eye_black ∧ under_tail_black∧belly_solid∧bill_black∧(bill_all_purpose)∧(wing_white)∧(underparts_black)∧ (back_black) ∧ (upper_tail_black) ∧ (head_plain) ∧ (breast_black) ∧ (throat_black) ∧ (bill_length_about_the_same_as_head) ∧ (...)*

Figure 8: Selection of training examples satisfying learned rules for CUB. For brevity, we drop some of the irrelevant concepts and replace them with (...).

Table 9: Rulebook for CelebA ($n_C = 1$) (seed 1).

| | |
|---|---|
| (1) | $Black\_Hair ←$ |
| (2) | $Black\_Hair ← 5\_o\_Clock\_Shadow$ |
| (3) | $Black\_Hair ← ¬5\_o\_Clock\_Shadow$ |
| (4) | $Male ←$ |
| (5) | $Male ← 5\_o\_Clock\_Shadow$ |
| (6) | $Male ← ¬5\_o\_Clock\_Shadow$ |
| (7) | $Wavy\_Hair ←$ |
| (8) | $Wavy\_Hair ← 5\_o\_Clock\_Shadow$ |
| (9) | $Wavy\_Hair ← ¬5\_o\_Clock\_Shadow$ |

Table 10: Rulebook for CelebA ($n_C = 12$).

| | |
|---|---|
| (1) | $Black\_Hair \leftarrow 5\_o\_Clock\_Shadow \land Arched\_Eyebrows \land Bags\_Under\_Eyes \land Bald \land Bangs \land Big\_Lips \land Big\_Nose \land Blond\_Hair \land Blurry \land Brown\_Hair \land Bushy\_Eyebrows \land Chubby$ |
| (2) | $Black\_Hair \leftarrow \neg 5\_o\_Clock\_Shadow \land \neg Bags\_Under\_Eyes \land \neg Bald \land \neg Big\_Lips \land \neg Big\_Nose \land \neg Bushy\_Eyebrows \land \neg Chubby$ |
| (3) | $Black\_Hair \leftarrow \neg Arched\_Eyebrows \land \neg Blond\_Hair \land \neg Blurry$ |
| (4) | $Male \leftarrow 5\_o\_Clock\_Shadow \land Arched\_Eyebrows \land Bags\_Under\_Eyes \land Bald \land Bangs \land Big\_Lips \land Big\_Nose \land Blond\_Hair \land Blurry \land Brown\_Hair \land Bushy\_Eyebrows \land Chubby$ |
| (5) | $Male \leftarrow \neg 5\_o\_Clock\_Shadow \land \neg Bags\_Under\_Eyes \land \neg Bald \land \neg Blond\_Hair \land \neg Blurry \land \neg Brown\_Hair \land \neg Chubby$ |
| (6) | $Male \leftarrow \neg Bald \land \neg Blond\_Hair$ |
| (7) | $Wavy\_Hair \leftarrow 5\_o\_Clock\_Shadow \land Arched\_Eyebrows \land Bags\_Under\_Eyes \land Bald \land Bangs \land Big\_Lips \land Big\_Nose \land Blond\_Hair \land Blurry \land Brown\_Hair \land Bushy\_Eyebrows \land Chubby$ |
| (8) | $Wavy\_Hair \leftarrow \neg 5\_o\_Clock\_Shadow \land \neg Bags\_Under\_Eyes \land \neg Bald \land \neg Big\_Nose \land \neg Bushy\_Eyebrows \land \neg Chubby$ |
| (9) | $Wavy\_Hair \leftarrow \neg Arched\_Eyebrows \land \neg Bald \land \neg Blond\_Hair \land \neg Blurry \land \neg Chubby$ |

Table 11: Rulebook for CelebA ($n_C = 37$).

| | |
|---|---|
| (1) | $Black\_Hair \leftarrow 5\_o\_Clock\_Shadow \land Arched\_Eyebrows \land Bags\_Under\_Eyes \land Bald \land \neg Big\_Lips \land Chubby \land \neg Double\_Chin \land \neg Eyeglasses \land \neg Gray\_Hair \land Heavy\_Makeup \land \neg Mouth\_Slightly\_Open \land \neg No\_Beard \land Oval\_Face \land Pale\_Skin \land \neg Pointy\_Nose \land Receding\_Hairline \land \neg Rosy\_Cheeks \land \neg Wearing\_Necklace \land \neg Wearing\_Necktie$ |
| (2) | $Black\_Hair \leftarrow 5\_o\_Clock\_Shadow \land Arched\_Eyebrows \land Bags\_Under\_Eyes \land \neg Bald \land \neg Big\_Lips \land Big\_Nose \land Blond\_Hair \land \neg Bushy\_Eyebrows \land Chubby \land \neg Double\_Chin \land Eyeglasses \land Goatee \land \neg Gray\_Hair \land Heavy\_Makeup \land \neg Mouth\_Slightly\_Open \land Mustache \land \neg Narrow\_Eyes \land \neg No\_Beard \land Oval\_Face \land Pale\_Skin \land \neg Pointy\_Nose \land Receding\_Hairline \land Rosy\_Cheeks \land Straight\_Hair \land Wearing\_Hat \land Wearing\_Lipstick \land Wearing\_Necklace \land \neg Wearing\_Necktie \land Attractive \land \neg Young$ |
| (3) | $Black\_Hair \leftarrow \neg Arched\_Eyebrows \land \neg Bangs \land \neg Blond\_Hair \land \neg Blurry \land \neg Heavy\_Makeup \land \neg Narrow\_Eyes \land \neg Pale\_Skin \land \neg Pointy\_Nose \land \neg Rosy\_Cheeks \land \neg Wearing\_Earrings \land \neg Wearing\_Necklace$ |
| (4) | $Black\_Hair \leftarrow \neg Bald \land \neg Big\_Lips \land \neg Blurry \land \neg Chubby \land \neg Double\_Chin \land \neg Eyeglasses \land \neg Goatee \land \neg Gray\_Hair \land \neg Mustache \land \neg Narrow\_Eyes \land \neg Pale\_Skin \land \neg Pointy\_Nose \land \neg Receding\_Hairline \land \neg Rosy\_Cheeks \land \neg Sideburns \land \neg Wearing\_Necklace \land \neg Wearing\_Necktie$ |
| (5) | $Male \leftarrow 5\_o\_Clock\_Shadow \land Arched\_Eyebrows \land Bags\_Under\_Eyes \land \neg Bald \land Big\_Lips \land Big\_Nose \land Blond\_Hair \land Blurry \land Bushy\_Eyebrows \land Chubby \land \neg Double\_Chin \land Eyeglasses \land Goatee \land \neg Gray\_Hair \land Heavy\_Makeup \land High\_Cheekbones \land Mouth\_Slightly\_Open \land Mustache \land \neg Narrow\_Eyes \land \neg No\_Beard \land Oval\_Face \land Pale\_Skin \land \neg Pointy\_Nose \land Receding\_Hairline \land \neg Rosy\_Cheeks \land Straight\_Hair \land Wearing\_Hat \land Wearing\_Lipstick \land Wearing\_Necklace \land \neg Wearing\_Necktie \land Attractive \land \neg Young$ |
| (6) | $Male \leftarrow 5\_o\_Clock\_Shadow \land Arched\_Eyebrows \land \neg Bags\_Under\_Eyes \land \neg Big\_Lips \land Chubby \land \neg Double\_Chin \land \neg Eyeglasses \land \neg Goatee \land \neg Gray\_Hair \land High\_Cheekbones \land No\_Beard \land \neg Pointy\_Nose \land Receding\_Hairline \land \neg Rosy\_Cheeks \land \neg Sideburns \land \neg Wearing\_Necklace \land \neg Wearing\_Necktie$ |
| (7) | $Male \leftarrow \neg 5\_o\_Clock\_Shadow \land Arched\_Eyebrows \land \neg Bags\_Under\_Eyes \land \neg Bald \land Bangs \land Big\_Lips \land Big\_Nose \land Brown\_Hair \land Bushy\_Eyebrows \land \neg Chubby \land \neg Double\_Chin \land Eyeglasses \land Goatee \land Heavy\_Makeup \land Narrow\_Eyes \land \neg No\_Beard \land Oval\_Face \land Pale\_Skin \land Receding\_Hairline \land Wearing\_Earrings \land \neg Wearing\_Lipstick \land \neg Wearing\_Necklace \land \neg Wearing\_Necktie \land \neg Young$ |
| (8) | $Male \leftarrow \neg Bald \land \neg Blond\_Hair \land \neg Blurry \land \neg Gray\_Hair \land \neg Mustache \land \neg Narrow\_Eyes \land \neg Pale\_Skin \land \neg Receding\_Hairline \land \neg Rosy\_Cheeks \land \neg Wearing\_Necklace$ |
| (9) | $Wavy\_Hair \leftarrow 5\_o\_Clock\_Shadow \land Arched\_Eyebrows \land \neg Bags\_Under\_Eyes \land Bangs \land Big\_Lips \land Brown\_Hair \land Bushy\_Eyebrows \land \neg Chubby \land \neg Double\_Chin \land Eyeglasses \land Goatee \land Heavy\_Makeup \land \neg No\_Beard \land \neg Pointy\_Nose \land Receding\_Hairline \land Rosy\_Cheeks \land Sideburns \land \neg Wearing\_Earrings \land \neg Wearing\_Necklace \land \neg Wearing\_Necktie \land \neg Young$ |
| (10) | $Wavy\_Hair \leftarrow 5\_o\_Clock\_Shadow \land Arched\_Eyebrows \land \neg Big\_Lips \land Blond\_Hair \land Blurry \land \neg Brown\_Hair \land \neg Bushy\_Eyebrows \land Chubby \land \neg Double\_Chin \land Eyeglasses \land \neg Gray\_Hair \land Heavy\_Makeup \land \neg Mustache \land \neg Narrow\_Eyes \land \neg No\_Beard \land \neg Pointy\_Nose \land Receding\_Hairline \land Straight\_Hair \land Wearing\_Lipstick \land \neg Wearing\_Necklace \land \neg Wearing\_Necktie \land Attractive \land \neg Young$ |
| (11) | $Wavy\_Hair \leftarrow \neg 5\_o\_Clock\_Shadow \land \neg Bags\_Under\_Eyes \land \neg Bald \land \neg Big\_Nose \land \neg Blurry \land \neg Bushy\_Eyebrows \land \neg Chubby \land \neg Double\_Chin \land \neg Eyeglasses \land \neg Goatee \land \neg Gray\_Hair \land \neg Mustache \land \neg Narrow\_Eyes \land \neg Pale\_Skin \land \neg Receding\_Hairline \land \neg Sideburns \land \neg Straight\_Hair \land \neg Wearing\_Necklace \land \neg Wearing\_Necktie$ |
| (12) | $Wavy\_Hair \leftarrow \neg Bald \land \neg Chubby \land \neg Mustache \land \neg Narrow\_Eyes \land \neg Pale\_Skin \land \neg Receding\_Hairline \land \neg Rosy\_Cheeks \land \neg Wearing\_Necklace$ |

## D   Code, licenses and resources

For our experiments, we implemented the models in Python 3.11.5 using open source libraries. This includes PyTorch v2.1.1 (BSD license) [48], PyTorch-Lightning v2.1.2 (Apache license 2.0), scikit-learn v1.3.0 (BSD license) [49] and xgboost v2.0.3 (Apache license 2.0). We used CUDA v12.4. Plots were made using Matplotlib v3.8.0 (BSD license) [50]. Our code is publicly available at `https://github.com/daviddebot/CMR` under the Apache License, Version 2.0.

All datasets we used are freely available on the web with licenses:

- MNIST - CC BY-SA 3.0 DEED,
- CEBaB - CC BY 4.0 DEED,
- CUB - MIT License [6],
- CelebA - The CelebA dataset is available for non-commercial research purposes only[7].

We will not further distribute them.

The experiments for MNIST+, MNIST+$^*$, C-MNIST, CelebA and CEBaB were run on a machine with an NVIDIA GeForce GTX 1080 Ti, Intel(R) Xeon(R) CPU E5-2630 v4 @ 2.20GHz with 128 GB RAM. The experiment for CUB and the fine-tuning of the BERT model used for the CEBaB embeddings were run on a machine with i7-10750H CPU, 2.60GHz × 12, GeForce RTX 2060 GPU with 16 GB RAM. Table 12 shows the estimated total computation time for a single run per experiment.

Table 12: Estimated total computation time for a single run of each experiment.

| Experiment | Time (hours) |
|---|---|
| CelebA | 3.7 |
| CUB | 1.8 |
| CEBaB | 0.1 |
| MNIST+ | 4.1 |
| MNIST+ rule int. | 0.2 |
| MNIST+$^*$ | 4.1 |
| C-MNIST | 4.4 |

---

[6]`https://huggingface.co/datasets/cassiekang/cub200_dataset`
[7]`https://mmlab.ie.cuhk.edu.hk/projects/CelebA.html`

