# OpenReview forum: "Interpretable Concept-Based Memory Reasoning"
_NeurIPS.cc/2024/Conference — NeurIPS 2024 poster_

### Official Review · Reviewer_VcGf · 2024-07-10

**Soundness:** 3
**Presentation:** 3
**Contribution:** 3
**Rating:** 6
**Confidence:** 4

**Summary:**

This paper presents an extension of Concept Bottleneck Models by modeling task predictions as neural selections over a learnable memory of rules, that is jointly learned during the training phase together with all the other model parameters. The resulting model is fully differentiable and can be exploited not only for its interpretability but also for the formal verification of specific properties. Experiments are conducted on a set of well-known benchmarks for CBMs, aiming to assess generalization, explainability and verifiability of the model.

**Strengths:**

+ Novel and interesting extension of Concept Bottleneck Models
+ Combination of neural and symbolic approaches for rule learning and concept prediction
+ Wide experimentation on several benchmarks
+ Clarity of presentation

**Weaknesses:**

- No comparison with other neural-symbolic systems (see questions and limitations below)

**Questions:**

* Why doesn't the graphical model in Figure 1 contain also the edge from c to r? The hypothesis here seems that c is independent from r given x and y?

* Wouldn't it be possibile to use a (probabilistic) symbolic system to learn the rulebook for the memory, since the concepts are fully supervised during training? What is the advantage of the proposed approach? I understand that the system is fully differentiable, but maybe at the cost of higher complexity?

**Limitations:**

-- The proposed approach could be compared against some neural-symbolic system or some probabilistic symbolic system (ProbLog, DeepProbLog, Probabilistic Soft Logic, maybe even Aleph) to be used in place of the rulebook learner. That would show better the advantage of the use of the memory component and the joint learning.

Typos and minor corrections:
- "i.e." -> "i.e.,"
- "boolean" -> "Boolean"

---

> ### Author Rebuttal · Authors · 2024-08-05
>
> We first want to sincerely thank the reviewer for taking the time to read and review our paper. We are pleased that the reviewer considered CMR an interesting and novel extension of CBMs, and considered the presentation clear.
>
> &nbsp;
>
> **Question: The proposed approach could be compared against some neural-symbolic system or some probabilistic symbolic system (ProbLog, DeepProbLog, Probabilistic Soft Logic, [...]**
>
> Thank you for this interesting remark. Since CMR is a probabilistic model rather than a system, **we can directly implement CMR in a system like DeepProbLog**, as DeepProbLog provides logical and probabilistic semantics, and can incorporate neural networks through the use of neural predicates. Moreover, as the bodies of our rules are essentially bodies of horn clauses, the probabilistic semantics of the selection of one among these horn clauses brings CMR close to the semantics of stochastic logic programs [1]), and therefore, to the neurosymbolic system DeepStochLog [2]. We chose to avoid using these explanations in the main text to keep the presentation and notation as simple as possible.
>
> For a direct empirical comparison with more "standard" neurosymbolic models defined in the mentioned systems, it's important to first note the **fundamental differences in datasets typically used for neurosymbolic models versus CBMs**. In the former, the knowledge (logic rules) on how to predict the task given the concepts is typically provided, while there are no concept labels. In the latter, the knowledge is missing, but concept labels are provided. For this reason, we make our comparison with other CBMs.
>
> **Question (continued): […] maybe even Aleph**
>
> In our experiments, we made the **conscious decision to compare with a CBM that uses a decision tree** as symbolic rule learner for task prediction, as opposed to a CBM that uses other rule learners such as ILP systems like Popper or Aleph, since ILP shines in the relational setting (as opposed to the tabular / propositional one of CBMs) [3].
>
> &nbsp;
>
> **Question: "Wouldn't it be possible to use a (probabilistic) symbolic system to learn the rulebook for the memory, since the concepts are fully supervised during training? What is the advantage of the proposed approach? I understand that the system is fully differentiable, but maybe at the cost of higher complexity?"**
>
> Thank you for this very interesting question! The integration with symbolic learners is definitely a very interesting and different direction of neurosymbolic learning that we are planning to investigate. Actually, **we have a preliminary indication of this potential with one of our experiments** (lines 293-299), where we showcase that manually adding rules to the memory can improve the model that is being learned regarding the interpretability of the learned rules. We mention this possibility of manually adding rules to the rulebook in lines 181-192, and these rules can come from human experts or indeed from symbolic learners.
>
> The **advantage of the end-to-end rule learning is that the joint optimization will make CMR learn rules that are accurate in combination with the selector**. Importantly, these rules need not necessarily be accurate on their own, but they need to be accurate when used with the selector. This can be seen by considering the proof for Theorem 4.1: while these rules do not form a universal binary classifier on their own, they do achieve this when used in conjunction with the selector.
>
> As a response to this question, we have made **an additional experiment where we extract the rules learned by a decision tree and use them in multiple ways in CMR, to show the advantage of the rule learning component**, as explained in the caption of Figure 2 in the attachment to the general rebuttal. We show the effect on accuracy of (1) completely swapping out the rule learning component with the pre-obtained rules from the decision tree, and (2) adding the pre-obtained rules while also keeping the rule learning component. We used a decision tree as rule learner.
>
> &nbsp;
>
> **Question: "Why doesn't the graphical model in Figure 1 also contain the edge from c to r? The hypothesis here seems that c is independent of r given x and y?"**
>
> In the graphical model, r represents the rule that is selected for evaluation. The task y depends on both r and c since y is the outcome of evaluating r using c. We indeed model c as conditionally independently of r given the input x. While modelling a dependency from c to r is certainly a reasonable design choice, we decided to not include this edge. This decision is based on the fact that the information present in the concepts is extracted from x, and the edge from x to r suffices to capture these possible dependencies.
>
> &nbsp;
>
> We also want to thank you for mentioning typos in the text; we will correct them.
>
> &nbsp;
>
> [1] Cussens, J. Parameter Estimation in Stochastic Logic Programs. Machine Learning. 2001.
>
> [2] Winters et al. DeepStochLog: Neural Stochastic Logic Programming. 2022.
>
> [3] Muggleton, De Raedt. Inductive Logic Programming: Theory and Methods. 1994.

---

> > ### Author Response · Authors · 2024-08-12
> >
> > Please let us know if you have any further questions or things we could clarify further. If not, we would appreciate if you could consider updating your review based on our replies.

---

> > > ### Comment · Reviewer_VcGf · 2024-08-12
> > > **Rebuttal**
> > >
> > > I thank the authors for their replies, which helped in clarifying some of my doubts. I understand some of the limitations pointed out by the other reviewers, but I liked the idea overall, and thus I will keep my (positive) score unchanged.

---

### Official Review · Reviewer_TwPt · 2024-07-12

**Soundness:** 1
**Presentation:** 2
**Contribution:** 2
**Rating:** 3
**Confidence:** 4

**Summary:**

Concept learning is a very important and current area of research. The paper is motivated from this perspective of neurosymbolic concept learning. However, there are a number of flaws. The paper refers to rules and "reasoning" informally; it does not define syntax and semantics neither does it define the reasoning process. This makes it very difficult to check the claims of verifiability and even to understand what exactly is meant by the notation used in the paper e.g. in the proof of Theorem 1 which is trivial. Claims are made of soundness, interpretability and even intervention which are not substantiated. The experimental results do not help clarify any of the issues. The rules provided are difficult to interpret and unrelated to the motivation of the paper, e.g. instead of concepts some rules refer to different noises.

**Strengths:**

Concept learning and neurosymbolic AI are relevant current themes.

**Weaknesses:**

The notation is not formalized and the experimental analysis is limited in scope.

**Questions:**

How do you handle the explosion of exceptions with the use of negation, e.g. Red and Not Square implies Apple. Why only Not Square? Why not also Not Triangular, Not Oval and the list of all the other properties that do not change? This is known as the frame problem.

Since the syntax of the logic is not specified, we don't know the difference between e.g. the left arrow and the right double arrow. Also, how is equality used in the language?

**Limitations:**

There are far too many claims to do with verification, interpretability, reasoning that are not substantiated, i.e. not formalized properly or backed by experimental results.

---

> ### Author Rebuttal · Authors · 2024-08-05
>
> We first want to thank the reviewer for taking the time to read and review our paper.
>
> &nbsp;
>
> **Question: "The rules provided are difficult to interpret and unrelated to the motivation of the paper, e.g. instead of concepts some rules refer to different noises."**
>
> **In all rules, we only refer to concepts.** We believe you refer to Table 2, where some of the concepts are "noise_g" and "noise_b". These are concepts, and in the caption of the table we state that "g" and "b" are abbreviations for "good" and "bad". In the CEBAB dataset, the task is to classify restaurant reviews as positive or negative, and these concepts denote whether the noise in the restaurant is good or bad. We mention this setting of CEBAB in Section 6.1 (lines 251-252).
>
> &nbsp;
>
> **Question: "Since the syntax of the logic is not specified, we don't know the difference between e.g. the left arrow and the right double arrow. Also, how is equality used in the language?"**
>
> All the connectives are standard logic connectives. The \leftarrow is used as assignment (or a computation direction), as stated in natural language in the sentence introducing the formula (line 129). We can definitely add a sentence to avoid confusion. The purely propositional logic expression for the task prediction is stated in Eq. 1, where only standard operators are used. The equality symbol is used as an indicator function for the role r_ij being positive (P), negative (N) or irrelevant (I). Such terms can be considered ground terms, avoiding exiting the propositional setting. We can change e.g. r_ij = P into r_ij^P to avoid using equality.
>
> &nbsp;
>
> **Question: "How do you handle the explosion of exceptions with the use of negation, e.g. Red and Not Square implies Apple. Why only Not Square? Why not also Not Triangular, Not Oval and the list of all the other properties that do not change? This is known as the frame problem."**
>
> **The frame problem is addressed by the loss we use in the paper** (Eq. 4). Our loss aims at learning rules that are as close as possible to the seen concept activations during training, given the limited number of rules of the model. We explain our reasoning behind this in Section 4.2 ("Interpretability"): we consider rules to be meaningful if they are prototypes of the data, which follows standard theories in cognitive science (line 160). Some other CBMs do suffer from the frame problem (e.g. DCR [1]) and the added regularization loss solves their issues.
>
> &nbsp;
>
> **Question: "There are far too many claims to do with verification, interpretability and reasoning that are not substantiated, i.e. not formalized properly or backed by experimental results."**
>
> We refer to the response to General Question 4.
>
> &nbsp;
>
> [1] Barbiero et al. Interpretable neural-symbolic concept reasoning. 2023.

---

> > ### Author Response · Authors · 2024-08-12
> >
> > Please let us know if you have any further questions or things we could clarify further. If not, we would appreciate if you could consider updating your review based on our replies.

---

> > > ### Comment · Area_Chair_wQMJ · 2024-08-13
> > >
> > > Dear Reviewer,
> > >
> > > given you were quite critical about the paper, I would appreciate if you could comment on the author's rebuttal, in light of the upcoming deadline.
> > >
> > > Thank you,
> > > Your AC

---

### Official Review · Reviewer_jRYi · 2024-07-13

**Soundness:** 3
**Presentation:** 3
**Contribution:** 3
**Rating:** 6
**Confidence:** 3

**Summary:**

The authors presented a novel framework to explain image classification models, specifically with an explainable-by-design deep model. This model is built to provide interpretability through discovering logic rules that matches ground truths. Intervention towards modifying the rules that changes the interplay among the concepts can be really helpful to incorporate the user preferences. The proposed methods does not much on accuracy-interpretability performances compared to state-of-the-art concept bottleneck models.

**Strengths:**

1) The proposed method discovers logic rules consisting of ground truth concepts that matches the true logic rules as working mechanism of a predictor model.
2) The concepts as well as the rules are allowed to intervene that incorporates expert opinion about the dataset specially when the model does not behave properly.
3) The properties behind model predictions and explanation can be verified before model deployment that shows the efficacy of the model in a specific scenario or deployment environment.
4) The experimental results prove the efficacy of the derived networks in terms of predictive performance measure, apart from generating nice explanations.

**Weaknesses:**

1) Line 49-52 explains some example rules for image classification task, which seems very simple. But, how are they going to work in practice for more complex scenarios? For example, an apple with a blue background can not decide the image not containing an apple just based on the concept 'blue' is active.

2) This method does not seem nicely scalable to datasets with complex scenarios (such as above) containing huge number of concepts and critical rule interactions. I would request the authors to comment on that.

3) How do you decide the number of optimal rules in the rulebook? Is it trying different numbers and choosing the number with the best accuracy performance? Or, is there any expert opinion involved?

4) How the correctness of the rules should be verified for more critical datasets? Should there be always a human to check the rules at the end?

**Questions:**

1) Did the authors check the time consumption for different datasets and architectures? While the rules are more useful for the experiments shown in the paper, it should not be hugely computationally expensive compared to the other concept based models.

**Limitations:**

The authors explained limitations, but missed explaining potential negative societal impact of this work. I would be happy to check author response on above comments and improve my score based on that.

---

> ### Author Rebuttal · Authors · 2024-08-05
>
> We first want to sincerely thank the reviewer for taking the time to read and review our paper.
>
> &nbsp;
>
> **Question: Lines 49-52 explain some example rules for an image classification task, which seem very simple. How are they going to work in practice for more complex scenarios? For example, one cannot decide that an apple with a blue background does not contain an apple just based on the concept 'blue' being active.**
>
> The provided example is intentionally simple to illustrate the basic mechanism, as in practice often more nuanced and specific concepts are used. For example, in the CUB dataset for bird classification, there is a concept denoting whether there is a "solid tail". Still, **while it is true that most CBMs' task accuracy heavily depends on the chosen concepts, CMR does not suffer from this problem.** This is due to the rule selector, which chooses which rule to apply for a given input; this allows even simple rules to make accurate predictions for complex scenarios. We also refer to what we stated in response to General Question 1.
>
> &nbsp;
>
> **Question: "How do you decide the number of rules in the rulebook?" Is it trying different numbers and choosing the number with the best accuracy performance? Or is there any expert opinion involved?"**
>
> As a consequence of Theorem 4.1, once the number of rules is 3 or larger, CMR can in principle achieve black-box accuracy (i.e. like a deep neural network) no matter the concepts. In response to this and one of your following questions, **we have added an experiment showcasing the robustness of CMR's accuracy to the number of rules**, for which we refer to the caption of Figure 1 in the attachment to the general response.
> **Rather than accuracy, setting the number of rules influences the specificity and granularity of the learned rules.** The more rules, the more fine-grained they will be. This also impacts on the intervention capabilities of the human: fine-grained rules may allow for very targeted modifications of the model's behaviour with rule interventions. While it is not required for the working of the model (i.e. for obtaining good accuracy and meaningful rules), human interaction can be needed for finding the preferred granularity.
>
> &nbsp;
>
> **Question: "How should the correctness of the rules be verified for more critical datasets? Should there be always a human to check the rules at the end?"**
>
> **The human should not necessarily go over all the rules, but could just automatically verify whether the model satisfies some criteria for their specific use case** (e.g. whether a constraint is satisfied). This is because rules serve a double scope: (1) they form an interpretable prediction system that a human can indeed inspect; (2) they form a formal logic system for prediction, that an automatic verifier (e.g. model checker) can use.
>
> &nbsp;
>
> **Question: How does the method scale to datasets with complex scenarios containing a huge number of concepts and critical rule interactions? Is CMR not too computationally expensive compared to other CBMs?**
>
> **As indicated in Theorem 5.1, making a task prediction using CMR is computationally linear in the number of concepts (like other CBMs) and rules.** However, we acknowledge that if the number of concepts and/or rules becomes excessively large, it is possible that CMR's prediction would be deemed too slow. For the dependency on the number of concepts, we refer to the answer of General Question 1, where we discuss that, in contrast to most CBMs, the number of concepts in CMR can be reduced without harming accuracy. For the dependency on the number of rules, we have shown with Theorem 4.1 and with the additional experiment of Figure 1 in the attachment to the general response that CMR can achieve the same accuracy with very few rules as with many rules. Therefore, reducing the number of rules can speed up CMR, with the trade-off being a potential decrease in interpretability, while accuracy remains robust. **So, while there is a linear increase in complexity, this is a reasonable price to pay for the added interpretability, especially since this complexity can be tuned as desired.**
>
> &nbsp;
>
> **Question: "The authors explained limitations, but missed explaining potential negative societal impact of this work."**
>
> Thank you for mentioning this. This remark corresponds to General Question 2, and we will make the changes mentioned in its answer.

---

> ### Comment · Reviewer_jRYi · 2024-08-12
> **Further comments**
>
> I would like to thank the authors for providing a detailed response. I have one more comment/clarification regarding accuracy results shown in Table 1. For some of the datasets, black-box models seem to perform worse than concept extraction based models. Why and when do you think that should happen?

---

> > ### Author Response · Authors · 2024-08-12
> >
> > Thank you for this interesting question. There are two dimensions to consider here:
> > 1) Concepts can either be or not be a bottleneck for the prediction task (i.e. they are not a bottleneck when the task can entirely be predicted from the concepts).
> > 2) Concept supervision is another source of information (and black-box models do not exploit it).
> >
> > The obtained results must be interpreted according to these two dimensions. The datasets for which this happens are MNIST+, MNIST+* and C-MNIST. In these datasets, **the concepts are completely _sufficient_ for the tasks**, meaning that the task can be predicted with 100% accuracy based on the ground truth concepts alone. For this reason, the concepts do not form a bottleneck for the model’s accuracy w.r.t. the black-box model (as they contain all information needed to perfectly predict the task), provided that concept accuracy is high. In these datasets, **concept accuracy is extremely high** (i.e. > 99%, Table 5), which then explains why the CBMs do at least as good. Moreover, they can do even better than the black-box model, as **the concept labels provide valuable information that the black-box model lacks**.

---

> > > ### Comment · Reviewer_jRYi · 2024-08-13
> > > **Reply to Authors**
> > >
> > > I thank the authors for providing clarifications to my questions/doubts and hence I am increasing my score.

---

### Official Review · Reviewer_ERBX · 2024-07-15

**Soundness:** 3
**Presentation:** 2
**Contribution:** 2
**Rating:** 6
**Confidence:** 3

**Summary:**

In this paper, the authors propose Concept-based Memory Reasoner (CMR), consisting of (a) a concept encoder, (b) a rule selector, and (b) a task-predictor. CMR is tested on a few different datasets-- easy (MNIST+, MNIST+∗, C-MNIST), medium (CEBAB), and hard (CelebA, CUB) for task and concept prediction accuracy, discovery of rules, verifiability of test behaviour, and the possibility of concept and rule-based interventions.

**Strengths:**

- CMR achieves similar or higher task accuracy on the range of datasets considered than existing concept bottleneck models considered on both complete and incomplete concept sets.

**Weaknesses:**

- What are the rules and where are they coming from in the rulebook? Also, how do we know that the rules being selected for an input are valid for the concepts present in it? I think the motivation behind selecting the rulebook and how they connect with the concepts can be explained better.
- What is an example of "rule 1" and how does it differ from the decoded rule in Figure 2?
- In the research questions posed in section 4, the authors mention they evaluate concept accuracy-- where are the ground truth concepts for this obtained from? Can you mention it somewhere?
- What do you mean by the CMR being "globally interpretable"-- how is the rule selector interpretable?
- Why are rules only seen as being conjunctive?
- In figure 3, why are CMR, CEM, and the black box model not much affected in terms of task accuracy with a varying number of concepts, whereas the other methods are? Also, how does the number of concepts on the x axis relate to the number of missing concepts? How many total concepts are there in CelebA?
- Why does CBM+linear achieve an accuracy of 0 on MNIST+ and MNIST+*?

Writing:

The writing of the paper can be heavily improved in terms of claims/remarks/adjectives peppered throughout the paper such as "CMRs are still foundational models" (how? how do they relate with foundation models?), "the first deep learning model
that is concept-based, globally interpretable and provably verifiable" etc.

Also, it is not really clear to me what exactly the rulebook and the concepts are and where they come from. If these things can be clarified, and the writing be made clearer, I am willing to reassess the paper and increase my score.

**Questions:**

- How does the use of a 2-layer MLP with ReLU to get the input embedding and 3 hidden layers with ReLU activation to get the concept prediction let the model remain globally interpretable?

**Limitations:**

The authors discuss the limitations and the positive societal impact of their work, though it would be good to see some negative societal impact listed as well.

---

> ### Author Rebuttal · Authors · 2024-08-05
>
> We want to sincerely thank the reviewer for taking the time to read and review our paper.
>
> &nbsp;
>
> **Question: "It is not really clear to me what exactly [...] the concepts are and where they come from. [For] concept accuracy, where are the ground truth concepts for this obtained from, and can you mention it somewhere?"**
>
> We refer to General Question 1 for the answer to this question. Additionally, some examples of concepts provided by the datasets can be found in the rules we provide (e.g. in Table 2). Some examples are "good food" in the context of restaurant reviews, "black wing" in the context of birds, and "bald" in the context of faces. Yes, we will mention this explicitly in the paper by adding the following sentence at line 255: "All these datasets come with full concept annotations."
>
> &nbsp;
>
> **Question: "What are the rules, and where are they coming from in the rulebook? What is an example of "rule 1" and how does it differ from the decoded rule in Figure 2?"**
>
> In Section 3.1.1 (lines 114-123), we explain that the model contains a rulebook, which is a set of learnable embeddings. Each embedding is decoded into a logic rule using a neural network. Therefore, **each embedding is a latent representation of a logic rule**. We will make this more clear by adding at line 123 the following sentence: "This way, each embedding in the rulebook is a latent representation of a logic rule." **Each rule is a conjunction of (possibly negated) concepts** (e.g. "red AND NOT soft -> apple"), which we mention in lines 91-92, and can therefore be evaluated using concepts to predict a specific class. In Figure 2, "rule 1" is such an embedding, and the decoded rule is its logical representation that can be evaluated using concepts.
>
> &nbsp;
>
> **Question: "Also, how do we know that the rules being selected for an input are valid for the concepts present in it?"**
>
> **The learning process guides the development of concepts, rules and the selection**. For each input, the selected rule is logically evaluated over the predicted concepts to provide the task prediction. Thus, the rules and the selection are automatically learned in such a way that the evaluation of the selected rule maximizes task accuracy, i.e. minimizes cross entropy on task ground truth labels (in addition to being prototypes of the data). In other words, rules are developed to ensure “validity” for the current task prediction. The training objective is given in Theorem 5.1 and Equation 4.
>
> &nbsp;
>
> **Question: "Why are rules only seen as being conjunctive?"**
>
> A single rule is a conjunction of possibly negated propositions (literals). In CMR, it represents the body of a horn clause, i.e. conjunctive_rule -> task, which is a possible alternative definition for the task. For example, in “red AND round” -> apple”, “red AND round” defines what an “apple” might be. **As multiple rules are allowed in the rulebook (multiple definitions for the task), the resulting language is very expressive** (the probabilistic semantics of the selection of one among multiple horn clauses brings CMR close to the semantics of stochastic logic programs [1]).
>
> &nbsp;
>
> **Question: "What do you mean by CMR being 'globally interpretable'? How is the rule selector interpretable? How does the use of [...] to get the concept prediction let the model remain globally interpretable?"**
>
> These questions correspond to General Question 3, and we refer to the answers given there.
>
> &nbsp;
>
> **Question: "In Figure 3, why are CMR, CEM, and the black-box model not much affected in terms of task accuracy with a varying number of concepts, whereas the other methods are?"**
>
> **For CMR and CEM, this is a key advantage over the remaining methods**, and we refer to what we stated in response to General Question 1. The black-box model does not use concepts for making predictions, as it is just a deep neural network; hence, its accuracy remains unaffected by the number of employed concepts and serves as a reference point for comparison with the concept-based models, which we mention in lines 266-267.
>
> &nbsp;
>
> **Question: "[In Figure 3,] how does the number of concepts on the x-axis relate to the number of missing concepts? How many total concepts are there in CelebA?"**
>
> The number of concepts completely to the right (37) is the total number of concepts available for the CelebA dataset, and the number of concepts on the x-axis is the number of concepts employed in each model. We mention this in lines 248-249, and we will make this clearer by changing the figure's caption to "Task accuracy on CelebA with varying numbers of _employed_ concepts."
>
> &nbsp;
>
> **Question: "Why does CBM+linear achieve an accuracy of 0 on MNIST+ and MNIST+*?"**
>
> The intuition behind this result is to be found in the choice of the subset accuracy metric, as task prediction is modelled as a multilabel classification problem and this introduced a strong class imbalance in MNIST+(*). As some of the tasks are clearly non-linear, the linear prediction has a very low chance to make all 19 tasks correct for any example.
>
> &nbsp;
>
> **Question: What do you mean with "CMRs are still foundational models"?**
>
> Thank you for asking this. With "foundational", we actually meant "fundamental", as CMR represents the first step towards a new class of interpretable CBMs that are globally interpretable and verifiable. We will replace the term as it better reflects what we meant.
>
> &nbsp;
>
> **Question: "The authors discuss the limitations and the positive societal impact of their work, though it would be good to see some negative societal impact listed as well."**
>
> Thank you for making this remark. This remark corresponds to General Question 2, and we will make the changes mentioned in its answer.
>
> &nbsp;
>
> [1] Cussens, J. Parameter Estimation in Stochastic Logic Programs. Machine Learning, 2001.

---

> > ### Author Response · Authors · 2024-08-12
> >
> > Please let us know if you have any further questions or things we could clarify further. If not, we would appreciate if you could consider updating your review based on our replies.

---

> > > ### Comment · Area_Chair_wQMJ · 2024-08-13
> > >
> > > Dear Reviewer,
> > >
> > > I would appreciate if you could comment on the author's rebuttal, in light of the upcoming deadline.
> > >
> > > Thank you,
> > > Your AC

---

> > > > ### Comment · Reviewer_ERBX · 2024-08-14
> > > > **reply to the authors**
> > > >
> > > > I thank the reviewers for addressing my comments in detail.
> > > > - I understand better the presence of the concepts' supervision and their global interpretability-- I still think the former (on supervision coming through concept annotations) can be clarified better in the paper and would advise the authors to do so.
> > > > - I thank the authors for including the negative societal impact of their paper.
> > > >
> > > > I would urge the authors to edit the writing of their paper to make it clear what information on concepts is a part of the problem, and am increasing my score to 6 based on the authors' response.

---

### Author Rebuttal · Authors · 2024-08-05

We first thank the reviewers for their insightful feedback. It has certainly improved the quality of our manuscript, and we hope we have been able to address all the raised concerns in this rebuttal. We reply to questions shared by multiple reviewers in this comment, and reply to specific questions in comments under their respective review.

&nbsp;

# Additional experiments

In response to some of the questions, we have performed two additional experiments (see the attached PDF for figures) with our model CMR.

- (@Rev-jRYi) **The first experiment shows the robustness of CMR's accuracy with respect to the number of rules it is allowed to learn.**

- (@Rev-VcGf) **The second experiment expands upon CMR's rule interventions, and also serves as an ablation study on the rule learning component.**

&nbsp;

# Answers to common questions

We paraphrase the shared questions and provide answers:

&nbsp;

**General Question 1 (@Rev-ERBX, @Rev-jRYi): "Where do the concepts and their supervision come from? Can the choice of concepts harm the accuracy of CMR and competitors?"**

- **The standard protocol in CBM literature is that concepts and their supervision are part of the problem (i.e. the dataset).** As the goal of such models is to be interpretable by an end user, concepts need to be designed as a communication language between the model and the users. Supervision needs to be provided as a form of alignment between user and model, as they need to assign the same meaning to the concepts.
- **Most CBMs have the significant limitation that their accuracy depends on the employed concepts, reducing performance w.r.t. deep neural networks** (black-box models). Therefore, decreasing the number of employed concepts is a trade-off, improving the computational efficiency, conciseness of the explanations, and effort from human annotators, but possibly harming accuracy and interpretability.
- **CMR can achieve black-box accuracy regardless of the employed concepts** (as a consequence of Theorem 4.1, and as shown empirically, e.g. in Figure 3) **_and_ CMR is globally interpretable** (as all decision rules are fully accessible, mentioned in lines 150-151). Thus, CMR removes the accuracy from the aforementioned trade-off: changing the number of concepts affects in general only its interpretability. While some other CBMs can achieve black-box accuracy regardless of the concepts, they lack global interpretability (e.g. CEM [1], lines 320-321).
- Therefore, CMR improves on the interpretability-accuracy trade-off w.r.t. other CBMs: (1) **among CBMs that can obtain black-box accuracy regardless of the concepts, CMR is the most interpretable, being the only globally interpretable one**, and (2) **among CBMs that are globally interpretable** (e.g. CBMs using logistic regression and CBMs using decision trees), **only CMR can obtain black-box accuracy regardless of the concepts.**

&nbsp;

**General Question 2 (@Rev-ERBX, @Rev-jRYi): "[...] Potential negative societal impacts would be welcome as well."**

As the human has direct access to the rules of CMR, the human can help resolve unfairness and bias by removing or changing such rules. However, the opposite is also possible: the human can add unfairness and bias this way. To provide a more nuanced view on the potential societal impacts, we will include this potential negative societal impact in the conclusion at line 346.

&nbsp;

**General Question 3 (@Rev-ERBX): "What do you mean by CMR being 'globally interpretable'?**"

Our interpretation of global interpretability follows the standard interpretation for CBMs. There are two terms involved: _interpretable_ and _global_. (1) CBMs are “intrinsically interpretable” models [4, 5] as they make explicit which human-interpretable concepts they use to make predictions. (2) A model is globally interpretable if the user "can interpret entire classes or sets of examples [...] and not just explain individual data inputs" [2, 3]. **CMR uses concepts to make its prediction, thus being _interpretable_, and, differently from most other CBMs, it allows inspecting the rules as descriptions of entire classes of examples and not on a “per-example basis”** (line 327)**, thus being _globally_ interpretable.**

&nbsp;

**General Question 4 (@Rev-TwPT): "Are the claims regarding interpretability, verifiability, intervenability and accuracy substantiated formally and/or empirically?"**

For the claim that CMR is a globally interpretable model, we refer to General Question 3. We evaluate the interpretability of the learned rules quantitively on 2 datasets (lines 284-288, Table 2) and qualitatively for all datasets (lines 289-291 and 707-746, Tables 2 and 5-11, Figures 5-6).

The claims made of having a verifiable task predictor follow from the fact that the task predictor has an equivalent logic formulation (Eq. 1). We explain how model properties can be verified (lines 194-204), and we have a verification experiment (lines 303-312).

We claim that with rule interventions during training, the human can directly shape the model that is being learned, which we explain in Section 4.2 (lines 181-192) and show with an experiment (Section 6.2.2, lines 293-299, Table 3). In Section 6.2.2 (lines 300-301), we also claim that CMR is responsive to concept interventions, shown with an experiment in Appendix C.3.1.

We have given a proof (line 142) that CMR is a universal binary approximator (Theorem 4.1). We show with our experiments (lines 270-282, Table 1, Figure 3) that this allows CMR to obtain similar accuracy to black-box models, regardless of the concepts.

&nbsp;

[1] Zarlenga et al. Concept embedding models. 2022.

[2] Kim et al. Interpretability beyond feature attribution: Quantitative testing with concept activation vectors. 2018.

[3] Doshi-Velez & Kim. Towards a rigorous science of interpretable machine learning. 2017.

[4] Molnar, Christoph. Interpretable machine learning. 2020.

[5] Koh et al. Concept bottleneck models. 2020.

---

### Decision · Program_Chairs · 2024-09-25

**Decision:**

Accept (poster)

**Comment:**

While most reviewers agree that the contribution (a concept-bottleneck model that includes a rule learning top layer) is valid and the empirical results support most of the claims made by the authors, the overall post-rebuttal rating is still somewhat borderline, with one reviewer being strongly against acceptance.

----

In order to get a better sense of the quality of the contribution, I have read the paper myself.  The paper is well written an presented.  The experimental setup is also okay, with one exception.  In fact, while I generally like the proposed architecture, I also think the paper has a serious issue that I feel obliged to voice here, broken down into three points:

- line 74: "we introduce Concept-based Memory Reasoner (CMR), the first deep learning model that is concept-based, globally interpretable and provably verifiable".  This statement is incorrect - many neuro-symbolc models comply with these desiderata; the authors actually reference a few of them in the paper - and should be either deleted or drastically toned down.

- In the Related Work, the authors correctly identify the strong ties between CMR and NeSy architectures, but ignore NeSy models that learn both the neural predicates and the rules.  A couple of examples are:

  Daniele et al. "Deep symbolic learning: Discovering symbols and rules from perceptions", 2022.

  Tang et al. "From perception to programs: regularize, overparameterize, and amortize", 2023.

There may be more recent works that I am unaware of.

- CBR is not compared against NeSy approaches that support rule learning.  It is unclear how DCR relates to them, performance wise (the DCR paper does not seem to acknowledge them at all, at least from my reading).   I think this substantially complicates assessing the novelty and significance of the contribution.  This aligns with reviewer VcGf opinion (although they were quite lenient about this issue).

----

UPDATE: I've thought about this paper for a while.  I think there is merit in the proposed approach.  The experimental setup is also quite convincing.  But, all in all, the paper should better position CBR against existing works from the NeSy AI camp. (The main differences being that NeSy approaches usually do not leverage supervision on the neural predicates; this would give CBR and edge.)

While I have been leaning towards rejection for a while, I think the paper contains enough new, useful and solid material to warrant acceptance, provided the authors address the reviewer's feedback (and my own) in the camera ready version.